



# Measurement report: Amino acids in fine and coarse atmospheric aerosol: concentrations, compositions, sources and possible bacterial degradation state

Ren-guo Zhu[1], Hua-Yun Xiao[2, 1*], Li Luo[1], Hongwei Xiao[1], Zequn Wen[3], Yuwen Zhu[1, 4], Xiaozheng Fang[1], Yuanyuan Pan[1], Zhenping Chen[1]

[1]Jiangxi Province Key Laboratory of the Causes and Control of Atmospheric Pollution, East China University of Technology, Nanchang 330013, China.

[2]School of Environmental Science and Engineering, Shanghai Jiao Tong University, Shanghai 200240, China

[3]Department of Earth Sciences, Faculty of Land Resource Engineering, Kunming University of Science and Technology, Kunming 650021, China

[4]School of Earth Sciences, East China University of Technology, Nanchang 330013, China;

*Corresponding author:* Hua-Yun Xiao (xiaohuayun@ecut.edu.cn)

**Abstracts.** The size distribution of amino acids (AAs) in atmospheric particles determines the atmospheric allergenicity, which may have deleterious effects on human health. This paper explores the use of compound-specific $\delta^{15}N$ patterns of hydrolyzed amino acid (HAA), $\delta^{15}N$ values of total hydrolyzed amino acid (THAA), degradation index (DI), and the variance within trophic AAs ($\sum V$) as markers to examine the sources and processing history of different sizes particle in the atmosphere. 2-weeks of daily aerosol samples from five sampling sites in the Nanchang area (Jiangxi Province, China) and samples of main emission sources of AAs in aerosols (biomass burning, soil and plants) were collected (Zhu et al., 2020). Here, we measured the concentrations and $\delta^{15}N$ values of each HAA in two size segregated aerosol particles (>2.5μm and PM2.5). Our results showed that the average concentrations of THAA in fine particles was nearly 6 times higher than that in coarse particles (p<0.0.1) and composition profiles of fine and coarse particles were quite different from each other. The $\delta^{15}N$ values of hydrolyzed glycine (Gly) in both fine (−1.0‰ to +20.3‰) and coarse particles (−0.8‰ to +15.7‰) exhibited wide ranges, but both fall within the ranges of Gly from biomass burning, soil and plant sources. Similarly, $\delta^{15}N$ values of THAA in both fine (+0.7‰ to +13.3‰) and coarse particles (−2.3‰ to +10.0‰) were typically in the range of THAA from these three emission sources. Moreover, the average difference in the $\delta^{15}N_{THAA}$ value between fine and coarse particles was smaller than 1.5‰. These results suggested that the sources of atmospheric AAs for fine and coarse particles might be similar, which are dominated by AAs from biomass burning, soil, and plant sources. Meanwhile, compared to fine particles, significantly lower DI values (*p*<0.05), "scattered" $\delta^{15}N$ distribution in Trophic-AA and higher $\sum V$ values (*p*<0.05) were observed in coarse particles. But the difference in $\delta^{15}N$ values of Source-AA (Gly, Ser, Phe and Lys) and THAA ($\delta^{15}N_{THAA}$) between coarse particles and fine particles was relatively small. It is likely that AAs in coarse particles have advanced bacterial degradation state compared to fine particles. Besides that, the significant increase in DI values and a decrease in $\sum V$ values for coarse particles were observed at the onset of the rainfall events (*p*<0.05). This implies that "fresh" AAs in coarse particles were mainly released at the onset of the precipitation and decayed swiftly.





## 1    Introduction

Recently, an increasing number of researchers highlight the importance of amino acids (AAs) in the atmosphere because AA is considered to be one of the most important compound classes of organic nitrogen compounds in atmosphere (Samy et al., 2013;Wedyan and Preston, 2008;Song et al., 2017;Zhang et al., 2002). Moreover, AAs are bioavailable and can be directly utilized by plant and soil communities. Its key role in atmosphere-biosphere nutrient cycling and global nitrogen cycle has aroused

greatly concern. Besides that, AAs and proteins are important constituents of allergenic bioaerosol. The distribution of AAs and proteins in different particle sizes will determine whether these compounds can reach the pulmonary alveoli and the allergy of aerosols. And the distribution of AAs associated with different particle sizes can help to trace the sources and transformation of atmospheric aerosols. However, detail information on the concentrations and mole composition profiles of AA distributed in different size

particle is still limited.

Compound-specific nitrogen isotope analysis of individual amino acids provide an opportunity to offer the key information on widely varied photochemical processes and origins of proteinaceous matter in the atmosphere. Nitrogen sources information and any possible nitrogen isotopic fractionation caused by transformation processes could be hold by the $\delta^{15}$N-AA pattern (Mccarthy et al., 2007;Bol et al., 2002).

At the same time, the $\delta^{15}$N value of total hydrolysable AA ($\delta^{15}$N$_{Avg-THAA}$), calculated as the average molar-weighted $\delta^{15}$N value of individual AA, has been used as a proxy for total protein $\delta^{15}$N value (Mccarthy et al., 2013). However, to our knowledge, no study has used the $\delta^{15}$N-AA pattern and $\delta^{15}$N$_{Avg-THAA}$ values to identify the sources of AAs distributed in different particle sizes.

It is generally accepted that AAs in aerosols are mainly controlled by abiotic photochemical aging

processes. On the contrary, the biological degradation of AAs in aerosols are neglected. This can be attributed to two factors. First, the sources and transformation pathways of protein matter and AAs in aerosols are highly complex (Wang et al., 2019;Zhu et al., 2020). Second, and the residence time of protein matter in aerosols is relatively short (Papastefanou, 2006). Admittedly, bacteria and fungi are ubiquitous and can be observed in all PM samples where people look for them, and this has been done

routinely for many decades (Bauer et al., 2002; Bowers et al., 2013; Huffman et al., 2013; Wei et al., 2016; Wei et al., 2019). *In-situ* bacterial degradation processes occurred in the aerosols and the cloud water was also observed (Amato et al., 2007; Husárová et al., 2011).Unfortunately, bacterial degradation of atmospheric AAs is rarely investigated, except for one study on marine aerosols by Wedyan and Preston (2008) and one study on precipitation by Yan et al. (2015). It is still unknown that whether

bacterial degradation play a role in the levels and compositions of AAs in different particle sizes.

In terrestrial, aquatic, and marine environment, the degradation state of organic materials (OM) is frequently characterized by DI based on the molar composition of amino acids (Dauwe and Middelburg, 1998;Wang et al., 2018;Dauwe et al., 1999). The higher DI values denote more "fresh" state of protein matter. Besides, a consensus has recently been reached on selective use of the $^{15}$N depleted or enriched

trophic AAs during bacterial heterotrophy processes can lead to large nitrogen isotopic fractionation in trophic AAs (McCarthy et al., 2004). Thus, substantial $\delta^{15}$N pattern shifts of trophic AAs can index bacterial heterotrophy processes. $\sum V$, defined as the average deviation in the $\delta^{15}$N values of the Tr-AA, has therefore been established to track the degree of bacterial degradation of AAs in marine and terrestrial





environment (Mccarthy et al., 2007;Philben et al., 2018;Yamaguchi et al., 2017).

In the present work, we sought to improve our understanding of AAs distributed in different sizes particle. We measured the concentrations and $\delta^{15}N$ values of each hydrolyzed amino acid in two size segregated aerosol particles (>2.5 μm and PM2.5) and main emission sources of AAs in aerosols collected in the Nanchang area (southeastern China). Furthermore, $\delta^{15}N$ values of Gly and THAA in fine and coarse particle were compared with those in main emission sources to identify the potential sources of fine and

coarse particles. In addition, the DI, $\sum V$ values and $\delta^{15}N$ values pattern of hydrolyzed AA in fine and coarse particles were analyzed to explore the possible bacterial degradation of AAs in fine and coarse particles.

## 2 Experimental section

### 2.1 Sample collection

Aerosol samples was collected at 5 locations included urban, town, suburban, airport and forest in Nanchang area (South China) from April 30, 2019 to May 13, 2019, using a high-volume air sampler (KC-1000, Qingdao Laoshan Electronic Instrument company, China) at a flow rate of $1.05 \pm 0.03$ m³ min⁻¹. The sampler allows to separate particles of different aerodynamic diameters in two stages with diameter (D) above 2.5 μm (coarse particles) and D≤2.5 μm (fine particles). Aerosol sampling was

conducted at the rooftop of the building in each site, about 10 meters above the ground except for the airport where the sampler was placed in a clear spot about 1000 meters away from the runway. The sampling time for each sample was from 5 p.m. to 4:30 p.m. of next day. More details on the sample collection are provided in Zhu et al. (2020).

Forest soil samples were collected at the top 10-cm of the evergreen broad-leaved forest soil in Nanchang

area (115.8 °E, 28.8 °N). Paddy soil samples were collected from the topmost 10-cm layer of rice cultivation soil (115.1 °E, 28.2 °N). Road soil was collected from highway topsoil (115.8 °E, 28.7 °N). Aerosols from straw burning were sampled by pumping into a high-volume air sampler (KC-1000, Qingdao Laoshan Electronic Instrument Company, China) from the funnel on the combustion furnace during July 2017. The combustion furnace is a domestic furnace widely used by local residents.

### 105 2.2 Analyses of the concentration and $\delta^{15}N$ value of individual HAA

For hydrolyzed AA analysis, samples were prepared using a modified version of Wang et al. (2019) and Ren et al. (2018). One-sixteenth of each fine aerosol filter (~80 m³ of air) or Two-seventh of each coarse aerosol filter (~366 m³ of air) was broken into small pieces and placed in a glass hydrolysis tube. Prior to the hydrolysis, 25 μL of ascorbic acid at a concentration of 20 μg μL⁻¹ (500 μg absolute) was added to

each filter sample. Then, 10mL and 6M Hydrochloric acid (HCl) was used to convert all of the combined AAs to free AAs. To avoid oxidation of AAs, the hydrolysis tube was flushed with nitrogen and tightly sealed before hydrolysis. The mixture was later placed in an oven at 110 ℃ for 24 h.

After cooling to room temperature, the hydrolyzed solution was dried with a stream of nitrogen and HCl was removed. The dried solution was then redissolved in 0.1 M HCl and purified by a cation exchange

column (Dowex 50W X 8H⁺, 200-400 mesh; Sigma-Aldrich, St Louis, MO, USA). Later, tert-



Butyldimethylsilyl (tBDMS) derivatives of HAAs were prepared following the method described by our previous study Zhu et al. (2018).

The concentrations of HAAs were analyzed using a gas chromatograph-mass spectrometer (GC-MS). The GC-MS instrument was composed of a Thermo Trace GC (Thermo Scientific, Bremen, Germany) connected into a Thermo ISQ QD single quadrupole MS. The single quadrupole MS was operated in electron impact ionization (70 eV electron energy) and full scan mode. The temperatures of the transfer line and ion source were 250 ℃ and 200 ℃, respectively. More details on quality assurance and control (recoveries, linearity, detection limits, quantitation limits, and corresponding effective limits in the aerosol samples of AAs), are provided in Zhu et al. (2020)

$\delta^{15}N$ values of AA-tert-butyl dimethylsilyl (tBDMS) derivatives were analyzed using a Thermo Trace GC (Thermo Scientific, Bremen, Germany) and a conflo IV interface (Thermo Scientific, Bremen, Germany) interfaced with a Thermo Delta V IRMS (Thermo Scientific, Bremen, Germany). The analytical precision (SD, n=3) of $\delta^{15}N$ was better than ±1.4‰. Moreover, AABA with known $\delta^{15}N$ value (-8.17‰±0.03‰) was added in each sample to check the accuracy of the isotope measurements. The analytical run was accepted when the differences of $\delta^{15}N$ values of AABA between GC- IRMS and EA-IRMS values were at most ±1.5‰. Each reported value is a mean of at least three $\delta^{15}N$ determinations. For more details of the analyses of HAA $\delta^{15}N$ values refer to our previous publication (Zhu et al., 2018). The concentrations and $\delta^{15}N$ value of Cys, Trp, Asn and Gln in HAAs could not be determined using this method because, under strong acidic condition, Cys and Trp is destroyed, and Asn and Gln are converted to Asp and Glu, respectively. The concentration and $\delta^{15}N$ value of hydrolysable Asp represents the sum of Asp and Asn; the concentration and $\delta^{15}N$ value of hydrolysable Glu represents the sum of Glu and Gln.

### 2.3 DI index

The degradation index (DI) was calculated using the formula Eq. (1) originally proposed by Dauwe et al. (1999):

$$DI = \sum_i (\frac{Var_i - Avg_i}{SD_i}) \times PC1_i \qquad (1)$$

where DI is the degradation index, Var is the mole% of the each individual HAA, $Avg_i$, $SD_i$ and $PC_i$ are the average mole% and standard deviation of each HAA in our data set, respectively, and PC1 loading of the amino acid $i$ obtained from principal component analysis (Table S1).

### 2.4 $\delta^{15}N$ values

The natural abundance of $^{15}N$ was calculated as $\delta^{15}N$ values in per mil (‰):

$$\delta^{15}N(‰ \text{ vs air}) = \left(\frac{R_{sample}}{R_{standard} - 1}\right) \times 1000 \qquad (2)$$

where R is the ratio of mass 29/mass 28

### 2.5 ∑V parameter

The ∑V parameter is defined as the average absolute deviation in the $\delta^{15}N$ values of the Trophic AA (including: Ala, Asp, Glu, Ile, Leu, and Pro) (Mccarthy et al., 2007). This parameter has been used as a proxy for the degree of heterotrophic resynthesis and calculated by Eq. (3):





$$\sum V = \frac{1}{n} \times \sum Abs(\chi_{AA}) \tag{3}$$

where $\chi_{AA}$ is defined as the deviation of the $\delta^{15}N$ of each trophic amino acid from the $\delta^{15}N$ of the mean of trophic amino acids ($\delta^{15}N$ AA- average $\delta^{15}N$ of Ala, Asp, Glu, Ile, Leu, and Pro), and n is the total number of trophic amino acids used in the calculation.

### 2.6 $\delta^{15}N_{THAA}$ values

The $\delta^{15}N$ values of total hydrolysable amino acids ($\delta^{15}N_{THAA}$) is calculated as the mole percent weighted sum of the $\delta^{15}N$ values of each individual HAA, following Eq. (4):

$$\delta^{15}N_{THAA} = \sum( \delta^{15}N_{THAA} \cdot \text{mol\%HAA}) \tag{4}$$

Where mol%HAA is the mole contribution of each HAA and $\delta^{15}N_{THAA}$ is the $\delta^{15}N$ value of individual HAA.

### 2.7 Statistics

All statistical analyses were performed using SPSS 16.0 (SPSS Science, USA). Graphs were generated using OriginPro 2018 (OriginLab Corporation, USA) and Sigmaplot 12.5 software (SPSS Science, USA). We performed a Two-way ANOVA for the concentration of THAA, the DI index, $\delta^{15}N_{THAA}$ values and $\sum V$ values, testing the effect of aerosol sizes, location, and their interaction. Tukey's Honestly Significant Differences (Tukey-HSD) test was used to evaluate which combinations of location and aerosol size were significantly different. Two-way ANOVA was also conducted for DI values, examining the effect of aerosol sizes, coefficients (obtained by using first principal component score or previous reported coefficients) and their interaction. The differences in $\delta^{15}N_{Gly}$ values for fine particles between 5 sampling locations were examined using the one-way analysis of variance (ANOVA) procedure, and compared using the Tukey-HSD test.

The exponential regression was analyzed to evaluate changes in DI index as a function of the concentration of THAA.

To test for changes in the concentration of THAA, DI index and $\sum V$ values following the rain events, a two-way ANOVA was performed, testing for effects of precipitation, aerosol sizes and their interactions. Tukey-HSD test was conducted to compare the significant difference. Changes in mol% of each HAAs concentrations following precipitation were tested for significance by using ANOVA procedure followed by a Tukey-HSD test to compare significant differences. For all tests, statistically significant differences were considered at $p < 0.05$.

## 3 Results

### 3.1 Concentrations of THAA

#### 3.1.1 Difference in the concentrations of THAA for fine and coarse particles

Fourteen hydrolyzed amino acids (Ala, Val, Leu, Ile, Pro, Gly, Ser, Thr, Phe, Asp, Glu, Lys, His and Tyr) were found in fine and coarse aerosol samples collected in Nanchang areas during spring 2019 (Fig. 1). The average concentrations of THAA in fine and coarse particles were 2542.9±1820.1 pmol m⁻³ and





434.0±722.6 pmol m$^{-3}$, respectively. The mean concentration of THAA for fine particles was nearly 6 times higher than that for coarse particles (p<0.01) (Fig. S1).

### 3.1.2 Concentrations of THAA at different locations

For fine particles, the average concentration of THAA in 5 sampling sites were significantly different (p<0.05), with the highest mean concentration of THAA in airport (3455.4±2203.7 pmol m$^{-3}$), followed by those in urban (2941.0±2443.5 pmol m$^{-3}$), forest (2730.2±1435.5 pmol m$^{-3}$) and town (2314.5±1211.7 pmol m$^{-3}$). The lowest THAA concentration occurred at suburban (1633.5±1087.2 pmol m$^{-3}$) (Fig. S1). However, for coarse particles, the difference in THAA concentrations between 5 sampling sites were not significant ($p$>0.05) (Fig. S1). The mean concentration of THAA in airport, urban, forest, town and suburban location was 540± 821.4 pmol m$^{-3}$, 230.9±300.8 pmol m$^{-3}$, 654.4±1152.4 pmol m$^{-3}$, 437.7±583.7 pmol m$^{-3}$ and 291.0±426.2 pmol m$^{-3}$, respectively.

### 3.1.3 Influence of precipitation on concentrations of THAA

Precipitation was observed to exert remarkable impacts on the concentrations of the THAA in fine particles. The average concentration of THAA in fine particles on rainfall days (1948.3±1546.8 pmol m$^{-3}$) was significantly lower than that measured on dry days (3137.5±1898.1 pmol m$^{-3}$) ($p < 0.05$), whereas the average concentrations of THAA in coarse particles displayed no significant changes during rain events ($p$>0.05) (Fig. 1 and Fig. S1). For coarse particles, the average concentrations of THAA on rainy and dry days was 660.3±947.4 pmol m$^{-3}$ and 212.2±266.8 pmol m$^{-3}$, respectively.

### 3.2 Amino acid mol% composition

#### 3.2.1 mol% composition profile of HAA in potential sources

The mol% composition of HAA were distinct in the biomass burning aerosols (straw burning), soil (road, paddy and forest soil), and plant sources (pine and straw) (Fig. 2). For straw burning, 11 hydrolyzed amino acids were detected in straw burning samples, including Ala, Val, Leu, Ile, Pro, Gly, Ser, Thr, Phe, Asp and Glu. Gly was the predominant HAA species from straw burning sources, contributed 45.8% of THAA pool. For soil sources, 12 hydrolyzed amino acids were detected, including Ala, Val, Leu, Ile, Pro, Gly, Ser, Thr, Phe, Glu, Lys and Tyr. Hydrophobic species (Ala, Val, Leu, and Ile) and neutral (Pro) were the most abundant HAA components in soil sources, together accounting for 81.8% of THAA pool in soil sources. For plant sources, 15 hydrolyzed amino acids were measured, including Ala, Val, Leu, Ile, Pro, Gly, Ser, Thr, Phe, Asp, Glu, Lys, His, Tyr and Met. Hydrophilic species (Lys and Asp), Hydrophobic species (Ala, Val, Leu, and Ile) and neutral (Pro) were the major HAA components, together accounting for 66.9% of THAA pool.

#### 3.2.2 Difference in composition profiles of HAA between fine and coarse particles

The composition profiles of HAA in fine and coarse particles during the whole campaign are shown in Fig. 3. The composition profiles of HAA in fine particles are quite different from those in coarse particles (Fig. 2 and Fig. 3). For fine particles, Gly, Pro, Leu and Glu were the four most abundant compounds, accounting for an average of 25.0 ±11.6%, 16.9 ±8.0%, 11.8 ±3.3% and 10.8 ±5.7%, respectively, of



the THAA pool.

For coarse particles, Pro were the most abundant THAA specie, with an average contribution of $63.3 \pm 31.1\%$ to the THAA pool. Leu, Ala and Val were the next most abundant species, each accounting for 6.6 -9.2% of the THAA pool, while other individual HAA was only minor component in coarse particles (Fig. 2 and Fig. 3).

### 3.2.3 Difference in composition profiles of HAA among different sampling locations

The HAA distribution among the different sampling locations for both fine and coarse particles appeared similar (Fig. 3).

### 3.2.4 Influence of precipitation on the composition profiles of HAA

It is worth noting that the influence of precipitation on the mole composition profile of HAA is different for the coarse and fine particles (Fig. 3). According to our previous study (Zhu et al., 2020) and the mol% composition of HAA observed in this study, hydrophobic (Ala, Val, Leu, and Ile), neutral (Pro) and

hydrophilic (Glu, Lys, and Asp) species are the major HAA components in natural sources (plant and soil sources). For fine particles, the contributions of HAAs species with high abundance in natural sources (including: Ala, Val, Leu, Ile, Pro, Glu, Lys, and Asp), increased from dry periods (65%) to rainfall periods (69%) (Fig. S2). Among these AAs, the percentage of Pro significantly increased from $14.1 \pm 6.2\%$ on dry days to $19.7 \pm 8.8\%$ on rainfall days ($p < 0.05$). There was no apparent trend in the percentage of

other individual HAAs for fine particles following the precipitation.

For coarse aerosol, the percentage of HAAs with high abundance in natural sources exhibited little changes from dry periods (averaged 94%) to rainfall periods (averaged 93%) (Fig. S2). However, the percentage composition of HAA in dry periods is quite different from that in rainy periods for coarse particles (Fig. 3). From dry periods to rainfall periods, the percentage of Pro in coarse particles

significantly decreased from $73.6 \pm 24.7\%$ to $52.7 \pm 33.6\%$ ($p < 0.05$), meanwhile the percentage of Ala, Val, Leu, Ile and Glu in coarse particles significantly increased ($p < 0.05$). These HAA species together accounted for 38.5% of the total THAA pool during dry periods, while during rainfall events this proportion was only 19.8%.

Besides that, compared to fine particles, the large variation in mole composition of THAA for coarse

particles was observed following rain events (Fig. 3). From dry periods to rainfall periods, the percentage changes of Pro for coarse particles (20.9%) was roughly 4 times greater than that for fine particles (5.6%). Similarly, from dry periods to rain periods, the increase in the percentage of Ala, Val, Leu, Ile and Glu in coarse particles is significantly greater than that in fine particles. For example, following the precipitation, Val in coarse particles increased by 3.5%, whereas Val in fine particles only increased by 0.3%.

Particularly, this steep decrease in the percentage of Pro and increase of other HAAs in coarse particles mainly occurred in the first day of rainfall events (Fig. 3).

### 3.3 DI values

Microbial degradation process could significantly modify the mole composition of protein amino acids (Dauwe et al., 1999). Accordingly, a quantitative degradation index (DI) has been developed based on



the mole composition of hydrolyzed amino acids pool (Yan et al., 2015).

### 3.3.1  PCA of HAA in fine and coarse aerosol particles

For calculation of DI values for fine and coarse particles, the first principal component score from principal component analysis (PCA) was applied to our own data (including Ala, Gly, Val, Leu, Ile, Pro, Ser, Thr, Phe, Asp, Glu, Lys, His and Tyr, except GABA), following the method described by Dauwe et
al. (1999).

The first principal component explained 38.3% of the variability, and the second principal component explained 20.9% (Table S1). Fig. 4a shows plots of the scores of the first and second principal components of fine and coarse particles in 5 sites. Components of fine and coarse particles could be roughly separated. The plots of the fine particles tended to cluster in the upper middle and right areas
(approximately -1.7 to +2.0, and -0.4 to 1.4 at first and second principal component scores, respectively). In contrast, the plots of the coarse particles tended to locate in the lower and left areas (approximately - 1.9 to 1.4, and -2.8 to +0.5 at first and second principal component scores, respectively). Fine and coarse particles were roughly distinguished by first and second principal component scores, suggesting that the first principal component score may also be designed as a degradation index of THAA in aerosols.

A plot of factor coefficients of each individual amino acid in the first and second principal components was examined to clarify the reasons for variation of the scores of fine and coarse particles (Fig. 4b). Based on this cross plot, 14 HAA species were divided into four groups. In Fig. 4b, Group 1 located in the lower right portion of the plot, included Val, Leu, Ile and Ala. Group 2, in the upper right of the plot, included Lys, Glu, Asp, Phe, Thr, Ser and Gly. Group 3, in the middle direction, included Tyr and His.
Group 4, in the left of the plot, included Pro. The principal component scores of atmospheric particles were affected by the relative abundance and the factor coefficient of each individual amino acid. The relative high principal component scores of fine particles in PC1 and PC2 were more affected by the high relative abundances of amino acids which has high factor coefficient (Group 1 and Group 2). In contrast, the relative low principal component scores of coarse particles in PC1 and PC2 were more affected by
the low relative abundances of amino acids which has low factor coefficient (Group 1 and Group 4).

Furthermore, DI values for fine particles showed positive correlation with percentage of HAA species in Group1 (e.g., Lys, Glu, Asp, Phe, Thr, Ser), but DI values for coarse particles were positively correlated to percentage of HAA species in Group 2 (e.g., Ala, Val, Leu and Ile) (Fig. 5), indicating the difference in composition profiles of HAA between fine and coarse particles may affected by the degradation
process.

### 3.3.2  Compared our calculating method with other works

This is the first report of the DI values for aerosol particles. We compared DI values obtained by our calculating method with those calculated by using the coefficients given in previous references (Dauwe et al., 1999;Yamashita and Tanoue, 2003). There is no significant difference between the DI values
calculated using the first principal component score and the DI values calculated using the coefficients given in the previous reference (Dauwe et al., 1999;Yamashita and Tanoue, 2003) (p>0.05) (Fig. S3), confirming our calculation method is reliable.


### 3.3.3 Correlation between DI and THAA

Plots of DI as a function of THAA concentration in both fine and coarse particles showed an exponential
relationship ($y=1067.4e^{-1.0x}$; $r=0.6$, $p<0.01$); that was, that at higher values of DI, concentrations of THAA
were higher, and vice versa (Fig. 6), indicating THAA for both fine and coarse aerosols probably
undergone bacterial degradation.

### 3.3.4 DI values of fine and coarse aerosol particles

DI values from literature data, where possible and DI values for fine and coarse aerosol particles are
shown in Fig. 7a and Fig. 8. Fine particles had significantly higher DI values than that of coarse particles
($p<0.05$) (Fig. 7a). The DI values for fine and coarse particles ranged from -0.3 to 1.4 (average=$0.6\pm0.4$)
and -1.8 to 1.4 (average=$-0.6\pm1.0$), respectively (Fig. 8). The DI values of fine particles were close to
those of "fresh" material. For instance, source materials (e.g., plankton, bacteria and sediment trap
material) and the precipitation in Uljin. On the contrary, the DI values of coarse particles were
comparable to those of surface soil, POM in coastal sediments and DOM in coastal area and precipitation
in Seoul, which were proved to be more degraded materials (Fig. 8).

### 3.3.5 DI values of aerosol particles at different locations

However, the differences in DI values were not significant among 5 sampling sites for both fine and
coarse particles ($p>0.05$) (Fig. S4). For fine particles, the average DI values in airport, urban, forest, town
and suburban location was $0.6\pm0.4$, $0.5\pm0.5$, $0.7\pm0.3$, $0.6\pm0.3$ and $0.7\pm0.2$, respectively. For coarse
particles, the mean DI values in airport, urban, forest, town and suburban location was $-0.5\pm0.9$, $-1.0\pm1.1$,
$-0.8\pm1.1$, $-0.3\pm1.1$ and $-0.5\pm1.1$, respectively.

### 3.3.6 Influence of precipitation on the DI values of aerosol particles

As exhibited in Fig. 7a, DI values of coarse aerosol particles were influenced by precipitation. For coarse
aerosol particles, a significant increase was found from dry (average=$-1.0\pm0.8$) to rain periods (average=$-0.3\pm1.1$) ($p<0.05$), whereas the DI values of fine particles during dry (average=$0.7\pm0.3$) and rain periods
(average=$0.6\pm0.4$) were not significantly different ($p>0.05$).

## 3.4 $\delta^{15}$N-AA patterns

### 3.4.1 $\delta^{15}$N-AA patterns of fine and coarse particles

The $\delta^{15}$N value for each individual AA measured and average amino acids in 5 sites were provided in
Fig. 9. The $\delta^{15}$N-AA pattern for most AA was similar between fine and coarse aerosol particles. Generally,
of the individual amino acids, Ala and Leu were enriched relative to $\delta^{15}N_{THAA}$, while Pro, Thr and Lys
exhibited depleted $\delta^{15}$N values to the $\delta^{15}N_{THAA}$ for both fine and coarse particles in 5 sites.

It is interesting to note that a substantial $\delta^{15}$N-AA shifts in trophic AA group was observed between fine
and coarse particles. Ala, Leu, Ile and Asp was $^{15}$N-enriched in coarse particles compared to fine particles,
whereas Pro in coarse particles was $^{15}$N-depleted than those in fine particles (Fig. 10). Clearly, there is
no uni-directional $^{15}$N depletion or enrichment of Trophic-AA was observed between fine and coarse



particle samples. The $\delta^{15}$N -AA distribution in the Trophic-AA group is more "scattered" in coarse particles than that in fine particles (Fig. 10). However, the difference in $\delta^{15}$N values of Source-AA between coarse particles and fine particles was relatively small except for Val. $\delta^{15}$N values of Gly, Ser, Phe and Lys measured in coarse particles are close to those measured in fine particles.

### 3.4.2 $\delta^{15}N_{Gly}$ and $\delta^{15}N_{THAA}$ values of local natural sources

Our previous study found that the $\delta^{15}$N value of Gly in PM2.5 can be used to trace the potential emission sources for aerosol AAs because the N isotope fractionation associated with Gly transformation in aerosol is relatively small (Zhu et al., 2020). The $\delta^{15}$N of protein AA ($\delta^{15}N_{THAA}$) has been served as a proxy for indicating the nutrient N in marine sediments (Batista et al., 2014). To trace the sources of fine and coarse particles, we measured the nitrogen isotopic compositions of hydrolyzed Gly and THAA sampled from main emission sources in the study areas, including biomass burning, soil and local plants (Fig. 10). The average $\delta^{15}$N value for hydrolyzed Gly from the biomass burning, soil, and plant sources was +15.6 ± 4.3‰, +3.0 ± 4.4‰, and −11.9±1.4‰, respectively, and the mean $\delta^{15}N_{THAA}$ value was +15.8 ± 4.5‰, +5.5 ± 2.2‰, and −0.0 ± 1.8‰, respectively.

### 3.4.3 $\delta^{15}N_{Gly}$ and $\delta^{15}N_{THAA}$ values in different size particles

In this study, to test $\delta^{15}N_{THAA}$ values of aerosol particles could also be used to trace the sources of aerosol particles, $\delta^{15}N_{THAA}$ values were compared with the $\delta^{15}N_{Gly}$ values. Since the concentration of hydrolyzed Gly is very low in coarse particles, a few the $\delta^{15}N_{Gly}$ values could be measured in coarse aerosol samples. Thus, only the $\delta^{15}N_{THAA}$ values of fine particles were compared with the $\delta^{15}N_{Gly}$ values of fine particles in the same sampling sites.

A remarkably consistent spatial-related trend was observed in $\delta^{15}N_{THAA}$ values and the $\delta^{15}$N values of hydrolyzed Gly (Fig. 11b and 11c). Both $\delta^{15}N_{Gly}$ values and the $\delta^{15}N_{THAA}$ values of fine particles in the urban and town locations showed more positive than those in suburban, airport and forest locations ($p<0.05$). Furthermore, the mean $\delta^{15}N_{THAA}$ value was not significantly different from the average $\delta^{15}$N value of hydrolyzed Gly in the 5 sampling locations ($p>0.05$), supporting $\delta^{15}N_{THAA}$ values of aerosols may also imprint the sources of atmospheric AAs.

The $\delta^{15}$N values of hydrolyzed Gly in fine and coarse particles exhibited wide ranges: −1.0‰ to +20.3‰ and −0.8‰ to +15.7‰, which fall within the ranges of biomass burning, soil, and plants sources. Similarly, both fine (+0.7‰ to +13.3‰) and coarse particles (−2.3‰ to +10.0‰) had the $\delta^{15}N_{THAA}$ value also typically in the range of these three main emission sources (Fig. 10). Therefore, it is likely that the main sources of atmospheric AAs for both fine and coarse particles were mainly biomass burning, soil, and plants.

In addition, the difference in $\delta^{15}N_{THAA}$ values were not significant for fine and coarse particles ($p>0.05$) (Fig. 11c) and the average offset of $\delta^{15}N_{THAA}$ value between fine and coarse particles was lower than 1.5 ± 1.7‰ in 5 sampling sites (Fig. 11a), further supporting the sources of AAs for fine and coarse aerosol particles may be similar.

### 3.4.4 $\delta^{15}N_{Gly}$ and $\delta^{15}N_{THAA}$ values of particles at different locations



The $\delta^{15}N_{Gly}$ values of fine particles was significantly different at 5 sampling sites (p<0.05). The average
$\delta^{15}N_{Gly}$ value of fine particles in urban (average=14.3 ± 8.5‰) and town (average=9.4 ± 4.2‰) were
more positive than that in suburban (average=6.7 ±4.3‰), airport (average= 6.9 ±5.3‰) and forest site
(average=6.5 ±5.0‰) (Fig. 11b). The significantly higher $\delta^{15}N_{Gly}$ values observed in the urban and town
locations suggested an increased contribution from biomass burning sources to Gly in fine particles at
these two locations.

Similar spatial variation trend in $\delta^{15}N_{THAA}$ values of fine and coarse particles among 5 sampling sites was
found. For fine particles, the highest $\delta^{15}N_{THAA}$ value of fine particles were observed in urban (average=9.4
± 2.5‰), town (average=8.4 ± 1.5‰), then in the suburban (average=5.4 ± 1.1‰), airport (average=5.9
± 2.8‰) and forest (average=5.7 ± 1.9‰) sites. For coarse particles, the most positive $\delta^{15}N_{THAA}$ value
were also occurred in urban (average=8.6 ± 0.9‰), town (average=7.0 ± 1.6‰), then in the suburban
(average=4.3 ± 3.4‰), airport (average=6.0 ± 3.1‰) and forest (average=5.4 ± 2.6‰) sites (Fig. 11c).
The more positive $\delta^{15}N_{THAA}$ values occurred in urban and town compared to other sampling sites for both
fine and coarse particles (p<0.05), indicating that atmospheric AAs for both fine and coarse particles in
urban and town were more influenced by biomass burning.

**3.5  ∑V**

∑V is defined as the average deviation in six Trophic-AA and has been proposed to reflect protein
resynthesis during microbial degradation processes (Mccarthy et al., 2007). Fig. 12 shows the ∑V values
measured in fine particles, coarse particles, and local natural sources, as well as ∑V values reported in
previous references.

**3.5.1  ∑V values in natural sources**

∑V values for main natural sources collected around the sampling sites were calculated. ∑V values for
local plants (needles of *Pinus massoniana* (Lamb.) and leaves of *Camphora officinarum*) ranged from
1.0‰ to 2.1‰, with a mean of 1.7±0.4‰ (Fig. 12). ∑V values in local soil (paddy soil, road soil and
forest soil) ranged from 1.4‰ to 2.1‰, with a mean of 1.7±0.3‰.

**3.5.2  Difference in ∑V between fine and coarse particles**

Overall, coarse particles had higher ∑V value (average = 3.6±1.5‰) than that of fine particles (p<0.05)
(Fig. 12). The mean ∑V value of fine particles in 5 sampling sites (average=2.4±1.1‰) was similar to or
slightly higher than that of plants and soil collected around sampling sites, phytoplankton (1.0‰) and
zooplankton (1.5‰) in marine (Mccarthy et al., 2007), needle (average=1.5±0.1‰), mosses
(average=1.1±0.02‰) and soil (average=1.4±0.1‰) measured in balsam fir forest (Philben et al., 2018),
and marine POM (average=2.3±0.7‰) (Batista et al., 2014;Mccarthy et al., 2007). In contrast, ∑V values
of coarse particles were equal to or even higher than those of more degraded materials, such as marine
dissolved organic matter reworked by bacterial heterotrophy (average =3.0±0.5‰) (Batista et al., 2014).

**3.5.3  ∑V value in aerosol particles at different locations**

The ∑V values of both fine and coarse particles were not significantly different among the 5 sampling





sites (p>0.05). The $\sum V$ values of fine particles in the urban, town, suburban, airport, and forest sites averaged 2.8±1.3‰, 2.3±0.7‰, 2.7±1.2‰, 2.4±1.0‰, and 1.9±1.0‰, respectively, and the $\sum V$ values of coarse particles averaged 2.5±1.0‰, 3.9±0.8‰, 3.3±1.1‰, 3.8±2.3‰, and 4.0±1.3‰ (Fig. 12).

### 3.5.4 Variations in $\sum V$ value of aerosol particles following rain events

Fig.7b shows the $\sum V$ values of fine and coarse particles during dry and rainy days. The $\sum V$ values of coarse aerosol particles were significantly affected by precipitation. From dry to rainy days, $\sum V$ values of coarse aerosol particles decreased from 4.5±1.5‰ to 3.0±1.3‰ ($p<0.05$). In contrast, the average $\sum V$ value of fine particles on dry and rainy days was identical (2.4±1.1‰).

### 4    Discussion

**4.1   Similar contribution sources of fine and coarse particles**

The sources of atmospheric proteinaceous matter are very complex. Primary biological aerosol particles (e.g, plants, soil, pollen, bacteria, fungi, spores and deris of living things), biomass burning, and agricultural activities are generally suggested to be the main contributing sources of atmospheric AAs (Matos et al., 2016;Mace et al., 2003). It is still unclear whether AAs fine and coarse particles influenced

by different sources. The detailed size-resolved investigation for the sources of atmospheric AAs is limited. Only Filippo et al. (2014) obtained very variable results for the size-segregated concentrations of atmospheric combined amino acids in the city Rome. In the warm season, highest concentration of CAAs distributed in the fine fraction, whereas, in the colder season, the increase distribution of CAAs in the coarse fractions was observed. This result could not provide conclusive evidence to define the origin

of atmospheric AAs in the different particle sizes.

With the development of stable N isotope technology, $\delta^{15}N$ values and $\delta^{15}N$ pattern has become effective tools to trace the sources of nitrogen compounds. Our previous study found that the $\delta^{15}N_{Gly}$ value in aerosol particles can be used to identify the potential emission sources for aerosol AAs. In this study, the $\delta^{15}N$ values of hydrolyzed Gly for both fine and coarse particles exhibited wide ranges but were typically

in the ranges of hydrolyzed Gly from biomass burning, soil and plant sources (Fig. 10). Therefore, it is likely that the main sources of atmospheric AAs for both fine and coarse particles were mainly biomass burning, soil, and plants.

Similarly, according to the $\delta^{15}N$ inventories of THAA in potential emission sources of atmospheric protein AA, $\delta^{15}N_{THAA}$ values for fine and coarse particles were also in the range of $\delta^{15}N_{THAA}$ values from

the three main contributing sources, which further demonstrated that AAs in both fine and coarse particle were affected by biomass burning, soil, and plants sources.

Moreover, the $\delta^{15}N$-HAA pattern of fine and coarse particles was remarkably consistent (Fig. 9 and 10). There is no significant difference in the $\delta^{15}N_{THAA}$ value between fine and coarse particles in each sampling sites ($p>0.05$) (Fig. 11c) and the average offset of $\delta^{15}N_{THAA}$ value between fine and coarse

particles was lower than 1.5‰ (Fig. 11a).Thus, it is suggested that the main sources of AAs in fine and coarse particles might be similar, all of which were influenced by biomass burning, soil, and plant sources.



### 4.2 Difference in degradation state of protein AA between fine and coarse aerosol particles

In this study, a huge difference was observed in the concentrations and mol% compositions of THAAs between fine and coarse particles (Fig. 1 and 3). As we discussed above, the sources of AAs in fine and coarse particles are similar, therefore this lager difference may be attributed to protein matter in fine and coarse undergoing different degrees of oxidation, nitration and oligomerization in the atmosphere (Liu et al., 2017;Wang et al., 2019;Song et al., 2017;Haan et al., 2009). Another possibility is that, bacterial degradation of AAs may contribute to this variation observed between fine and coarse particles. Interestingly, by using bacterial markers (DI, $\delta^{15}$N-AA distribution and $\sum$V), we found that AAs in fine and coarse particles may be degraded by bacteria to different degrees.

The degradation index (DI) value has been developed to assess degradation state of atmospheric matter solely derived from amino acids (e.g., protein materials) (Yan et al., 2015). Relatively high DI values denote a relatively "fresh" matter. In this study, the positive correlation between the concentrations of THAA and DI values for both fine and coarse particles in the atmosphere is established in this study (Fig. 6). Furthermore, for fine particles, DI exhibited positive correlation with the percentage of Lys, Glu, Asp, Phe, Thr, Ser, but for coarse particles, DI showed positive correlation with the percentage of different HAA species (e.g., Ala, Val, Leu and Ile) (Fig. 5). Clearly, concentrations of THAAs and composition profiles of HAA in aerosols may be related to microorganism-induced degradation processes. In compared to fine particles, the coarse particles had significantly lower THAA concentrations (Fig. S1) and DI values (Fig. 7). Moreover, compared with the DI values reported by previous studies, the DI values of coarse particles were comparable to those of more degraded materials (e.g., soil and POM in hemi pelagic sediments), while the DI values of fine particles were close to those of "fresh" material (e.g., plankton, bacteria and sediment trap material) (Fig. 8), implying that AAs in coarse particles may undergo more bacterial degradation than fine particles. Our result is also comparable to that observed in precipitation at Uljin and Seoul (Yan et al., 2015). The DI values measured in coarse particles are closer to those observed in Seoul, where is believed to have more advanced bacterial degradation than Uljin.

Previous studies proposed that a substantial $\delta^{15}$N-AA shifts in trophic AA group may indicate bacterial heterotrophy has occurred and those new resynthesized protein has become an important part of protein material measured (Mccarthy et al., 2007;Calleja et al., 2013). In this study, Ala and Ile were $^{15}$N-enriched in coarse particles compared to fine particles. Similarly, recent study also observed the strong $\delta^{15}$N shift of Ala and Ile is accompanied by the processes of bacterial heterotrophy (Calleja et al., 2013;Mccarthy et al., 2007). Moreover, there is no uni-directional depletion or enrichment of $\delta^{15}$N -AA distribution in the Trophic-AA group between fine and coarse particle samples. The $\delta^{15}$N -AA distribution in the Trophic-AA group is more "scattered" in coarse particles than that in fine particles. Specifically, in coarse particles, Ala and Leu were $^{15}$N-enriched and Pro was $^{15}$N- depleted than those in fine particles (Fig. 10). This "scattered" characteristic of $\delta^{15}$N-AA distribution in Tr-AA group of coarse particles may be due to the nitrogen fractionation occurred in microbial consumers selectively using Trophic-AA.

$\sum$V could reflect the increasing trend of "scatting" $\delta^{15}$N-Trophic AA pattern related to bacterial resynthesis (Calleja et al., 2013;Yamaguchi et al., 2017). Significantly increasing $\sum$V values indicate organic material are more degraded by heterotrophic bacteria (Batista et al., 2014). In this study, the mean $\sum V$ value of fine particles was similar to or slightly higher than that measured in "fresh" materials



(Mccarthy et al., 2007;Philben et al., 2018;Batista et al., 2014), while $\sum V$ values of coarse particles were equal to or even higher than those of more degraded materials (Fig. 12). Accordingly, the significant higher values of $\sum V$ measured in coarse particles than that in fine particles ($p<0.05$) (Fig. 5) may also

imply more bacterial heterotrophic resynthesis occurred in coarse particles.

Despite the uncertainties surrounding oxidation, nitration and oligomerization of AAs in the atmosphere, main observations remain that the difference in $\delta^{15}N$ values of Source-AA (Gly, Ser, Phe and Lys) and total hydrolysable amino acids ($\delta^{15}N_{THAA}$) between coarse particles and fine particles was relatively small (Fig. 10). The average offset of $\delta^{15}N_{THAA}$ value between fine and coarse particles was lower than 1.5‰

(Fig. 11a). These results appear to contrast with what one might expect for AAs in either sizes particles undergo particularly more photochemical transformation than the other. Therefore, significantly lower DI values, "scattered" characteristic of $\delta^{15}N$ distribution in Tr-AA and higher $\sum V$ values observed in coarse particles in this study provide evidence that the difference in the THAA concentration and mol% composition distribution between fine and coarse particles may be related to AAs in coarse particles have

stronger bacterial degradation state than those in fine particles.

### 4.3   Release of coarse "Fresh" bioparticles at the onset of the rainfall

A tight relationship between atmospheric bioaerosols and precipitation has been found by previous studies (Huffman et al., 2013;Yue et al., 2016). Since biological sources contain a large abundance of AAs (Ren et al., 2018), HAAs in aerosols can be used as tracer compounds to indicate the release of

biological sources during precipitation. However, detailed size-resolved and time-resolved observation for the release of bioparticles initiated by precipitation are spare and the degradation state of different sizes bioparticles has never been examined.

It is expected that the concentrations of individual AAs in aerosol were assumed to decrease during rainfall events because of the high scavenging ratio of AAs in aerosol (Gorzelska and Galloway, 1990).

In this study, form rain to dry periods, the concentrations of THAA for fine particles decreased ($p<0.05$) (Fig. S1), but the concentration of THAAs for coarse particles displayed not significant change ($p>0.05$) (Fig. S1). Similar variation trends of different size particles following the precipitation were also observed by (Huffman et al., 2013). They also found the steep increase of coarse particles while low concentrations of fluorescent bioparticles and total aerosol particles were found in fine particles during

the precipitation, suggesting the new released AAs during the precipitation are mainly distributed in coarse particles.

In addition, the large variations in the percentage of some HAA species (e.g., Pro, Ala, Val, Leu, Ile and Glu) were observed in coarse particles following the rainfall events, which imply the states of coarse particles measured during rain periods were different from the ones measured during dry periods (Fig.

3). This conclusion also supported by the variation of DI and $\sum V$ values for coarse particles following rain events. From dry to rain periods, DI values in coarse aerosol particles were significant increased ($p<0.05$) but the $\sum V$ value was significantly decreased ($p < 0.05$) (Fig. 7), suggesting more fresh AAs in coarse particles were released during rain events, whereas, on dry days AAs in coarse particles were more degraded.

Furthermore, we observed an obviously temporal variations of the concentration and mol% composition



of HAA for coarse particles during the precipitation. At the onset of every rainfall events, the highest concentration of THAAs in coarse particles was observed. However, as the rainfall continued, the concentration of THAAs decreased rapidly. This temporal variation trend can attributed to the active release of these fresh biological aerosols caused by droplets splashing on porous medium are much stronger at the onset of rainfall events but with the rainfall continued, bioparticles in coarse particles were suppressed by rain scavenging (Joung and Buie, 2015;Huffman et al., 2013;Yue et al., 2016). Moreover, the mol% composition of HAA in coarse particles measured at the onset of every rainfall events were significantly different from that observed in the continued stage of rainfall events. The mol% composition of HAA in the continued stage of rainfall events were similar to that observed on dry days (Fig. 2). As we discussed above, AAs in coarse particles on dry days were more degraded. Therefore, we can conclude that those "fresh" protein matters are prone to release at the onset of the rainfall events and decayed swiftly with the precipitation continued.

## 5    Conclusions

2-weeks of daily aerosol samples in two size (>2.5μm and PM2.5) were measured. The significant difference was observed in the characteristics and distributions of AAs between fine and coarse particles. The concentrations of THAA in fine particles were significantly higher than that in coarse particles. A huge difference was also observed in the mol% compositions of THAAs between fine and coarse particle. For fine particles, Gly, Pro, Leu and Glu were the four most abundant compounds of the THAA pool, whereas for coarse particles, Pro were the most abundant THAA specie and Leu, Ala and Val were the next most abundant species.

Similar $\delta^{15}$N-HAA pattern, closer $\delta^{15}N_{THAA}$ values and small offset of $\delta^{15}N_{THAA}$ value observed between fine and coarse particles indicates that AAs in fine and coarse particles might have the same sources. Moreover, the $\delta^{15}$N values of hydrolyzed Gly and THAA for both fine and coarse fall within the ranges of those measured in biomass burning, soil and plant sources suggested that atmospheric AAs for both fine and coarse particles were affected by these three emission sources.

The difference in $\delta^{15}$N values of Source-AA and THAA between coarse particles and fine particles were small, implying AAs in fine and coarse particles may undergo the same degree of photochemical transformation. Using bacterial marker, we found that AAs in coarse particles may undergo more bacterial degradation than fine particles, which is supported by lower DI values, "scattered" $\delta^{15}$N distribution in the Trophic-AA group and higher $\sum V$ values observed in coarse particles compared to fine particles.

By comparing the variation of DI and $\sum$V values for coarse particles following the precipitation, we propose that "fresh" protein matters distributed in coarse fraction are prone to release at the onset of the rainfall events.

*Author contributions*. Ren-guo, Zhu., Zequn Wen and Yuwen Zhu designed the experiments, performed analyses, and analyzed the data. Hua-Yun Xiao were the principal investigators of the project that supported this work. All the authors have helped in the discussion of the results and collaborated in writing this article.



*Acknowledgements.* This work was supported by the National Natural Science Foundation of China (Grant No.41425014 and 41463007), Key Laboratory Project of Jiangxi Province (20171BCD40010) and Two 1000 Talents Plan Project of Jiangxi Province (S2018CQKJ0755). We are thankful to Global Weather and Climate Information Network ([http://www.weatherandclimate.info/](http://www.weatherandclimate.info/)) for providing
meteorological parameters include temperature (T) and relative humidity (RH), during the sampling period.

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

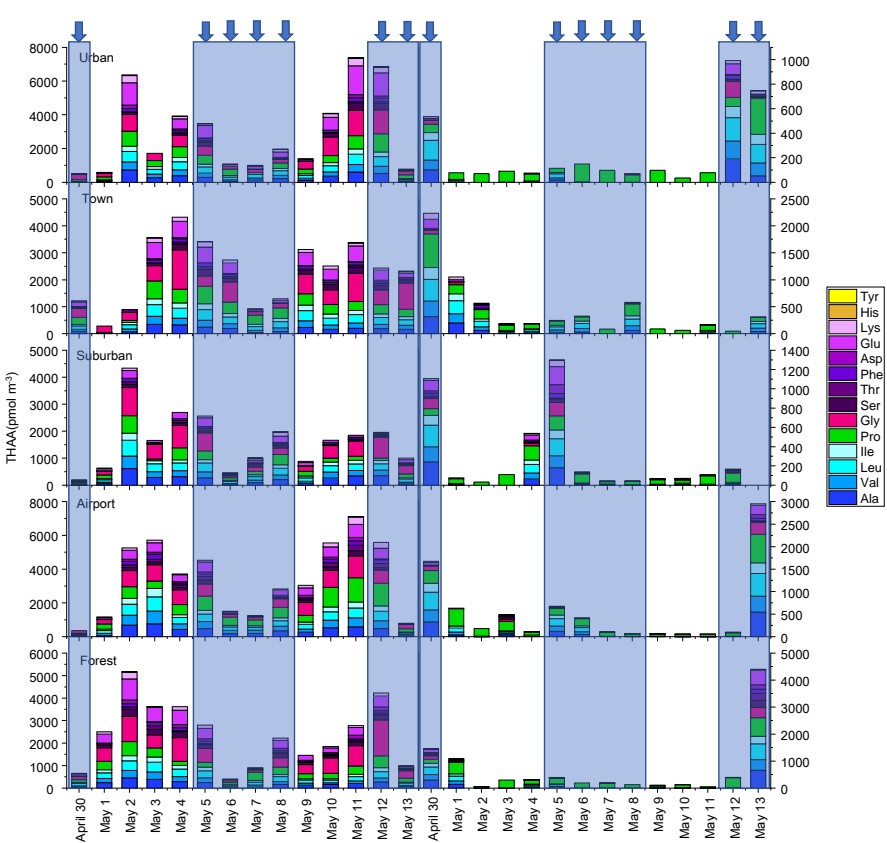

Figure1. Concentrations of hydrolyzed amino acids for fine and coarse particles in urban, town, suburban, airport and forest sites during 14 consecutive sampling days. The concentrations of HAAs for each sample were normalized for the total volume of air sampled. The blue arrow and shallow represent precipitation.

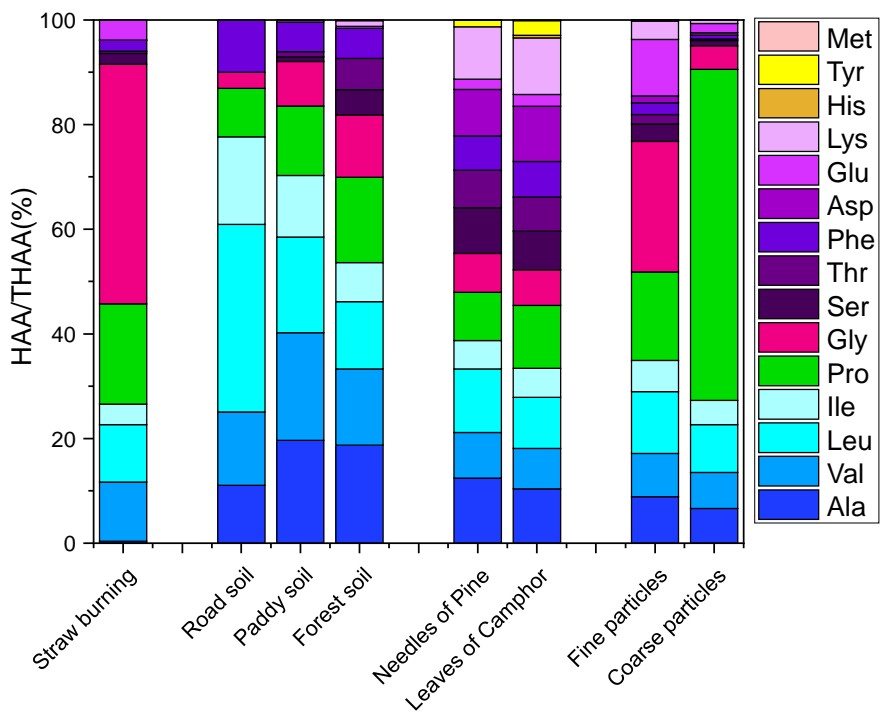

Figure 2. Average percentage composition of each hydrolyzed amino acids (% of THAA) in biomass burning (straw burning), soil (road, paddy and forest soil), plant sources (needles of *Pinus massoniana* (Lamb.) and leaves of *Camphora officinarum*), and fine and coarse aerosol particles.



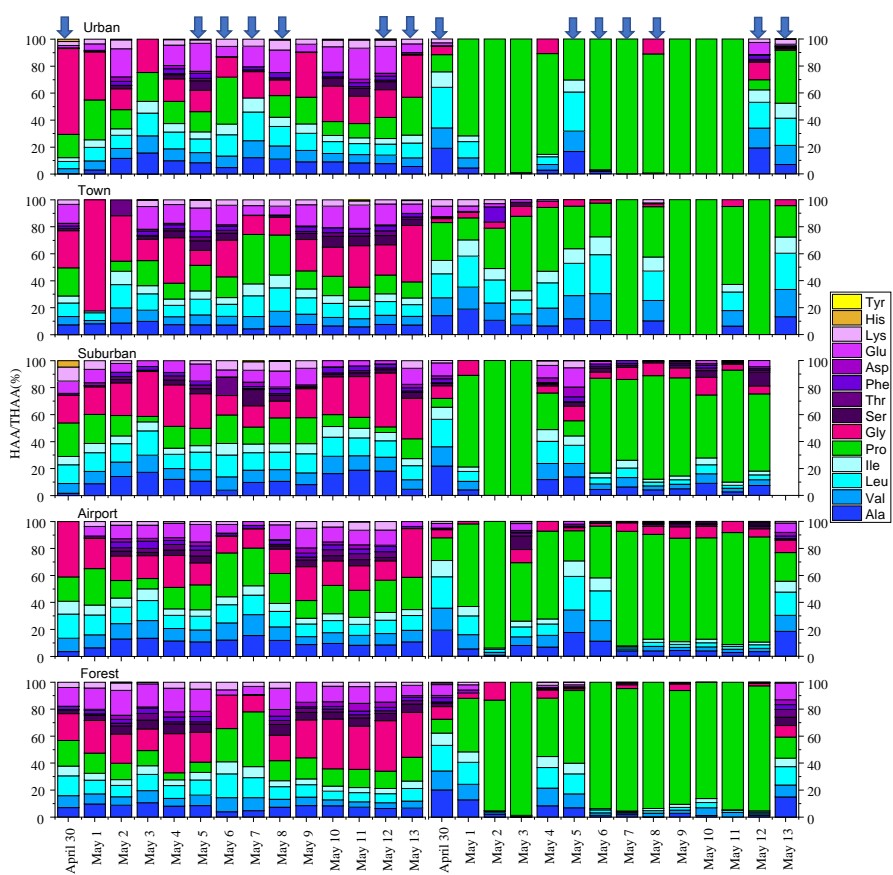

Figure 3. Percentage composition of each hydrolyzed amino acids (% of THAA) for fine and coarse aerosol particles in urban, town, suburban, airport and forest sites during 14 consecutive sampling days. The blue arrow represent precipitation.

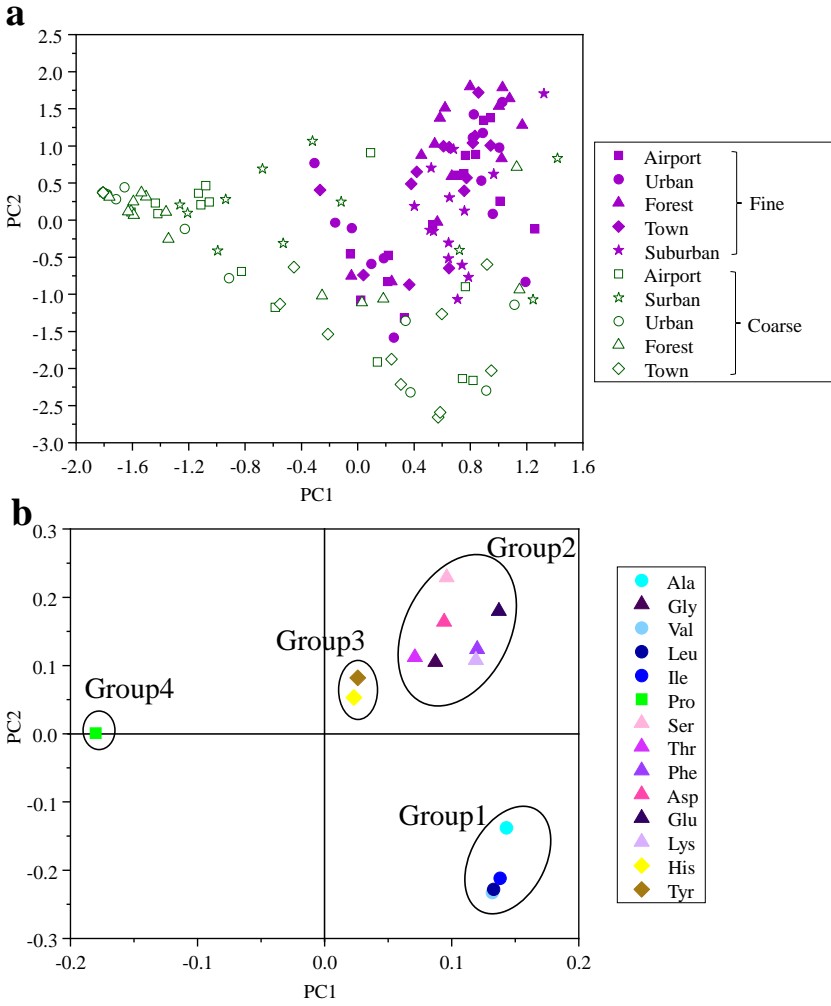

Figure 4. (a) Cross plot of the first and second component scores of PCA based on percentage composition (mol%) of hydrolyzed amino acid for fine and coarse particles. (b) Cross plot of factor coefficients of the first and second principal components of PCA. The lines enclosing each group of amino acid are arbitrarily drawn.

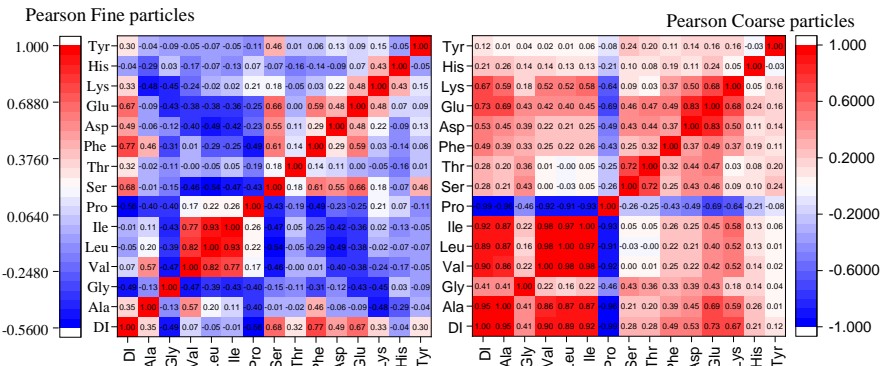

Figure 5. Correlation heatmap and associated significance of DI values with mol% percentage of hydrolyzed amino acid for fine and coarse particles. The color scale indicates negative/positive correlation.



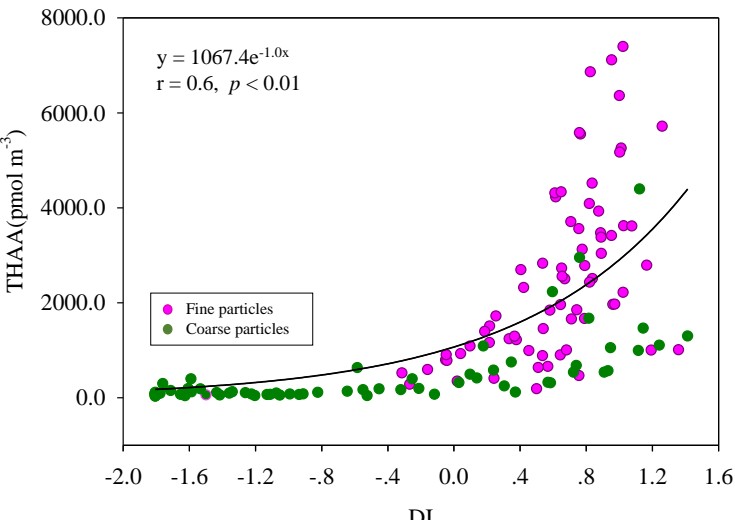

Figure 6. Correlations between concentrations of total hydrolyzed amino acid and DI values for fine and coarse particles.

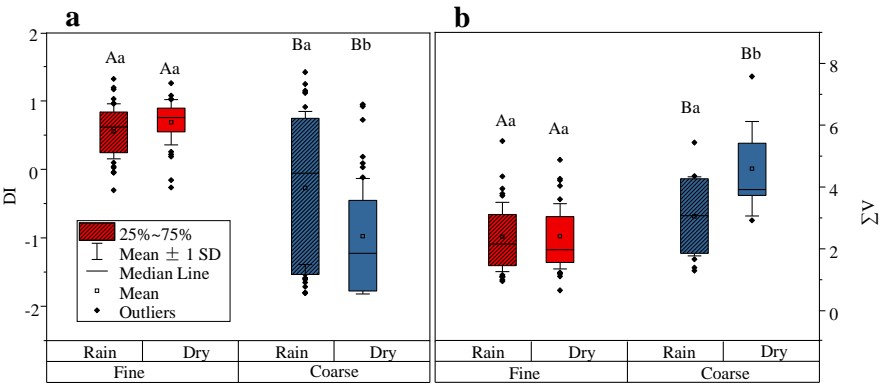

Figure 7. DI values (a) and (b) ∑V for fine (red box) and coarse (blue box) particles. The box encloses 50% of the data, the whisker is standard deviation of the data, the horizontal bar is the median, solid circles are outliers. The differences in means were statistically significant (two-way ANOVA, $p < 0.05$). Different uppercase letters denote means found to be statistically different (Tukey-HSD test) between fine and coarse particles. Different lower case letters denote means found to be statistically different (Tukey-HSD test) between rainy and dry days.





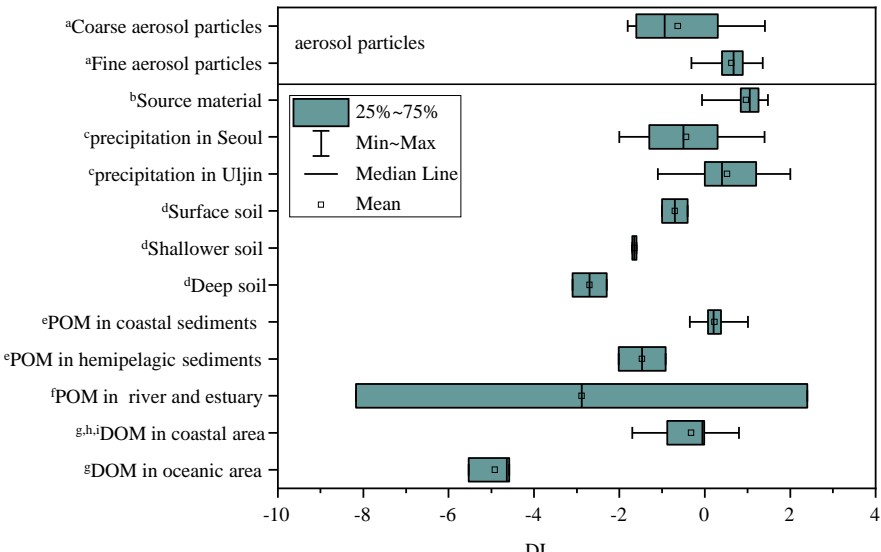

Figure 8. DI values of fine and coarse particles in comparison to other studies. a: this study. b: source materials including phytoplankton, bacteria, zooplankton and sediment trap material from Dauwe et al., 1999. c: Yan et al., 2015. d: Philben et al., 2015. e: particle organic matter from Mccarthy et al., 2007. f: Wang et al., 2018. g: Yamashita and Tanoue, 2003. h: Chen et al., 2016. i: Ji et al., 2019.

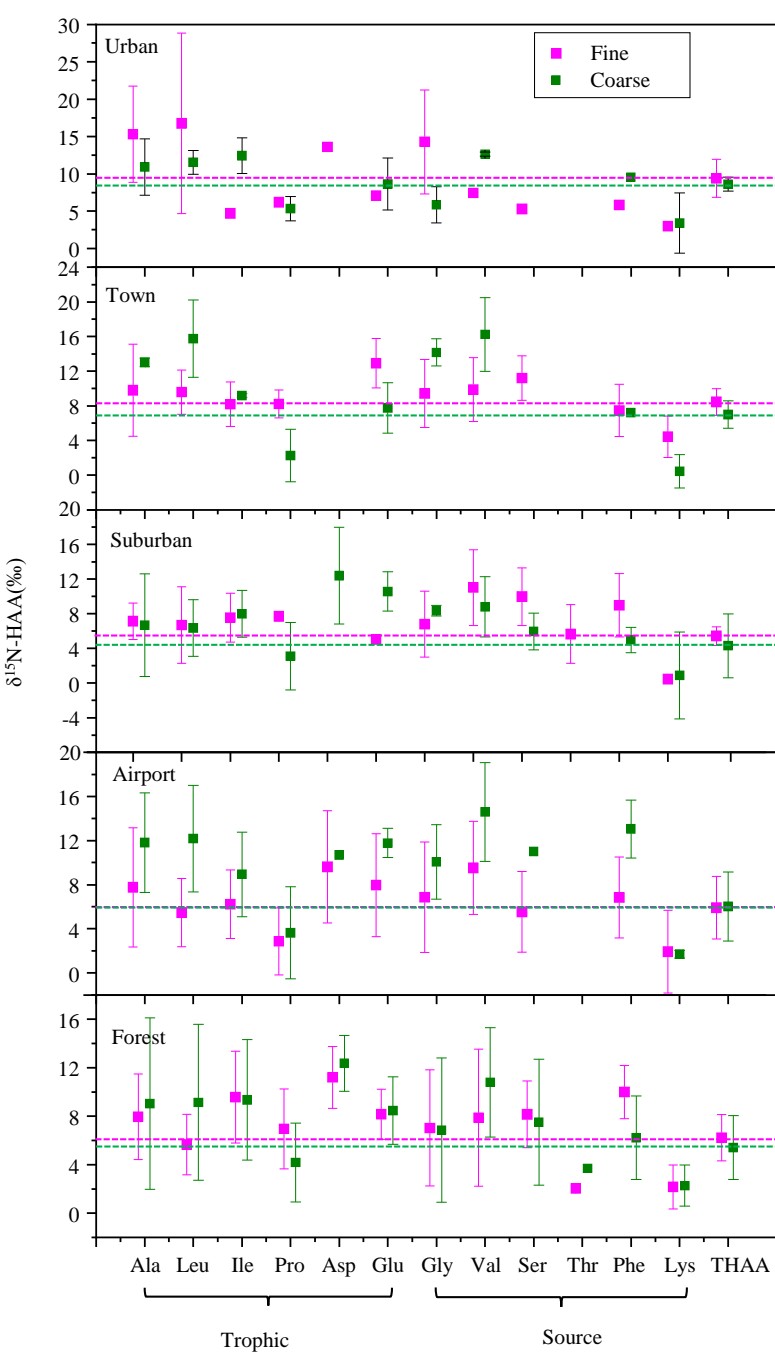

Figure 9. δ¹⁵N-HAA patterns of fine and coarse aerosol particles in urban, town, suburban, airport and forest sites.





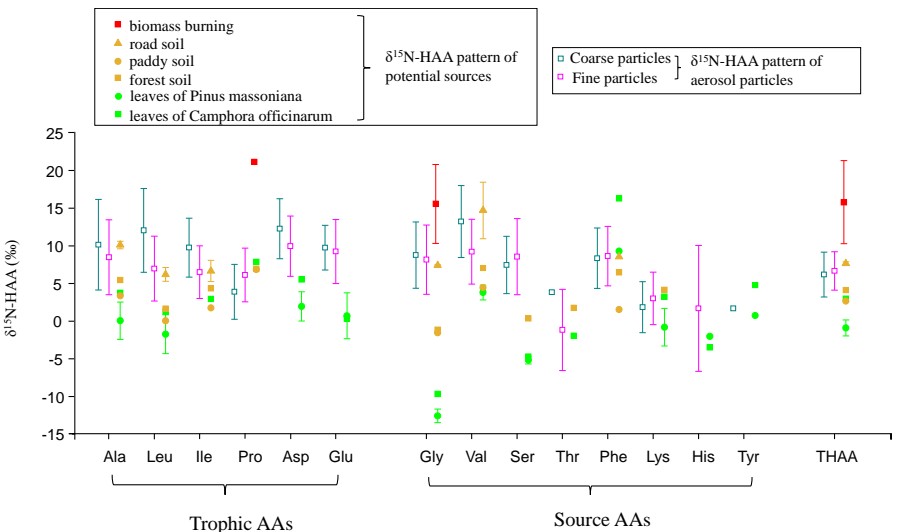

Figure 10. Comparison of δ¹⁵N-HAA patterns of fine and coarse aerosol particles with that of potential local sources.

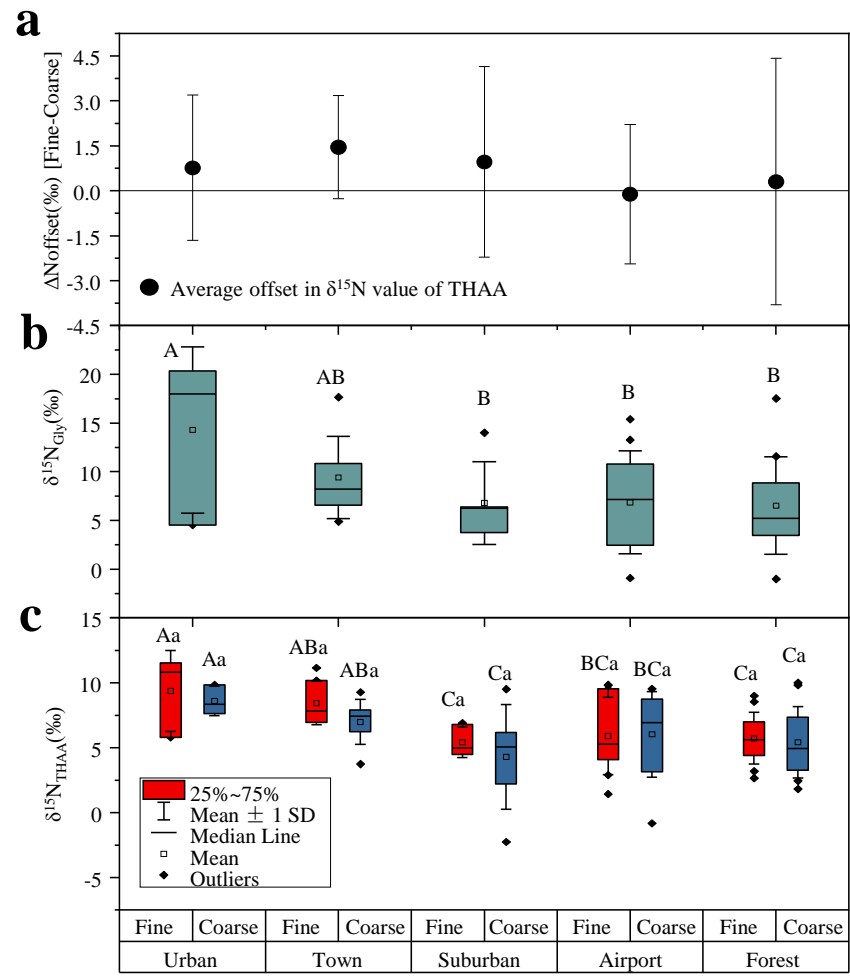

Figure 11. (a) The Offset of $\delta^{15}N_{THAA}$ values between fine and coarse particles; (b) The $\delta^{15}N_{Gly}$ values of fine particles; (c) The $\delta^{15}N_{THAA}$ values of fine and coarse particles in urban, town, suburban, airport and forest sites. Different uppercase letters denote means found to be statistically different (Tukey-HSD test) between sites. Different lower case letters denote a significant difference between fine and coarse particles. The error bars in (a) indicate the standard deviation.

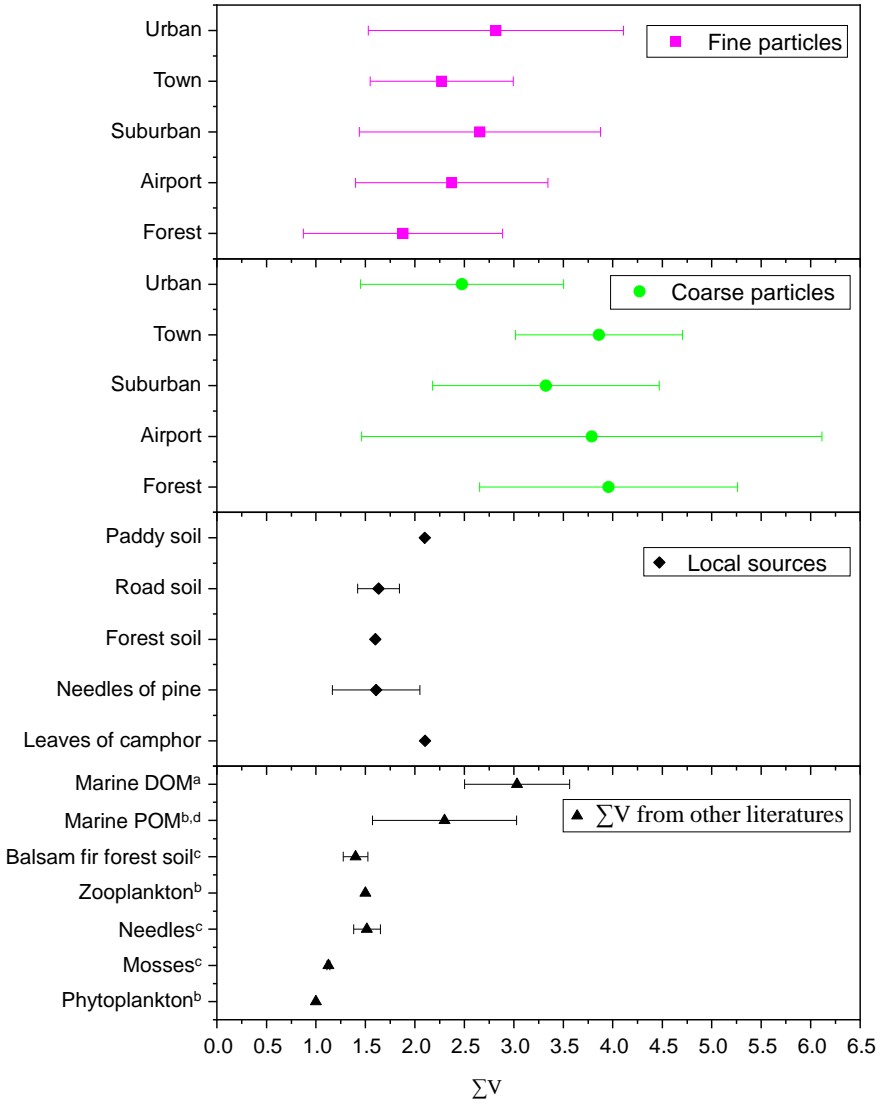

Figure 12. $\sum V$ values for fine and coarse particles in comparison to local natural sources and other studies. a: Calleja et al., 2013. b: Mccarthy et al., 2007. c: Philben et al., 2018. d: Batista et al., 2014.