# Peer review of "Measurement report: Hydrolyzed Amino acids in fine and coarse atmospheric aerosol in Nanchang, China: concentrations, compositions, sources and possible bacterial degradation state"

_Atmospheric Chemistry and Physics, 2020_

## Referee Comment (RC1) · Anonymous Referee #1 · 19 Aug 2020

The manuscript represents an important contribution to the study of atmospheric amino acids. These important compounds are recently better investigated, and the scientific community are step by step understanding their importance to investigate sources and processes in the atmosphere. For this reason, I think that this paper should be recommended for publication after addressing the issues listed below. In general, a very important dataset is present in this paper, but I found the manuscript, and in particular the results, quite hard to follow them. I suggest the authors to simplify the results, for example, reducing the number of sections and to link each section to help the readers.

[Figure]

General comments

Abstract. In the abstract the use of acronyms is inappropriate as well as it is dissuaded to insert the references.

Line 70. I think that you have to better introduce the degradation index to help the reader. I saw its explanation in section 2.3 but some details have to be introduced also in the introduction.

Section 2.1. I think that you have to add some information about the type of filter used and the cleaning procedure of this filter. You have to add the reference but I think that you have to insert this information in the main manuscript.

Section 3.2.3. This part is too short to be one section and I suggest to add this sentence to another section.

Line 295. I don't understand why you use PC1 as coefficient. This principal component clearly distinguishes the fine and the coarse particles. I think that this point should be clarified in the manuscript.

Section 3.3.4. I think that you have to define the meaning of DI values, also considering previous published results. You have to define the threshold when bacterial degradation occurred.

I don't like so much this fragmentation of the section. This is only my opinion, but I think that this fragmentation produces to lose the thread. You have the sections with 4-5 lines.

In the conclusion, you affirm that "The difference in $\delta15N$ values of Source-AA and THAA between coarse particles and fine particles were small," but one of the main aim of the manuscript is the follows: "$\delta15N$ values of Gly and THAA in fine and coarse particle were compared with those in main emission sources to identify the potential sources of fine and coarse particles.". So is the conclusion that $\delta15N$ values are not good tracers to define the sources?

Specific comments

Lines 42-43. I suggest rephrasing this part because I think that English form is not correct. For example you repeated "compound".

Line 43. I suggest to insert this reference because it summarized very well the state of knowledge in the 2016: "Matos, João TV, Regina MBO Duarte, and Armando C. Duarte. "Challenges in the identification and characterization of free amino acids and proteinaceous compounds in atmospheric aerosols: a critical review." TrAC Trends in Analytical Chemistry 75 (2016): 97-107."

Line 45. Here a reference is needed.

Line 50. I suggest you this paper where the particle size distribution of free amino acids is investigated until nano dimension: "Barbaro, et al. "Characterization of the water soluble fraction in ultrafine, fine, and coarse atmospheric aerosol." Science of The Total Environment 658 (2019): 1423-1439.".

Line 67-69. I think that you should also add the investigation of Kuznetsova et al. "Kuznetsova, M., Lee, C., Aller, J., 2005. Characterization of the proteinaceous matter in marine aerosols. Mar. Chem. 96, 359e377. https://doi.org/10.1016/j.marchem.2005.03.007

Line 114. Have you verified the recovery of amino acids from the cationic cation exchange column? Figure S1. Please add (F) and (C) in the caption after fine and coarse. Change "blue" with "green" because I saw green the coarse particles.

Lines 186-187 and in other sections of manuscript. Please consider to significant figures. For example, "2542.9±1820.1 pmol m-3" should be 2542±1820 or the best way is 3±2 nmol m-3. I found the same mistake in the % values.

Lines 421. Please consider that the combined amino acids were investigated also in the Arctic region, considering also the particle size distribution. Feltracco, et al. "Free and combined L-and D-amino acids in Arctic aerosol." Chemosphere 220 (2019): 412-

421.

Lines 430-432. You have completely skipped the marine contribution. Several studies conducted by prof. Leck (Leck and Bigg, 2005a, 2005b; Bigg, 2007; Bigg and Leck, 2008) demonstrated the sea emission of PBAP. Combined amino acids is surely one of the main component of PBAP.

Technical correction Line 23. Please remove one point from (p<0.0.1). Line 80. Please change as "particle sizes" Line 90. Change "was" with "were" Line 105. Please introduce the acronym HAA

---

## Author Comment (AC1) · 25 Aug 2020

Dear Reviewer:

Thank you for your letter and comments concerning our manuscript acp-2020-534 entitled "Measurement report: Amino acids in fine and coarse atmospheric aerosol: concentrations, compositions, sources and possible bacterial degradation state". Those comments are all valuable and very helpful improving our paper. We accept your suggestions and revise the manuscript according to the points as follows. Particularly, we

have reduced the number of results section.

Abstract. In the abstract the use of acronyms is inappropriate as well as it is dissuaded to insert the references. Answer: Thank you for your suggestion. Since the samples analyzed in this paper were obtained in the same sampling campaign of Zhu et al. (2020), according to editor requirement, we referenced the Zhu et al. 2020 and clearly stated that the samples are the same in both papers but the data used in this manuscript were not analyzed in Zhu et al., (2020). Besides that, we try our best to reduce the use of acronyms in the abstract. Gly, Ser, Phe and Lys have changed to the full name in the abstract. But we still preserved some acronyms, including amino acids (AAs), hydrolyzed amino acid (HAA), $\delta$15N values of total hydrolyzed amino acid ($\delta$15NTHAA), degradation index (DI), and the variance within trophic AAs (). These acronyms occurred several times in the abstract (at least three times) and the full name of these acronyms are long. To reduce the length of the abstract and improve the readability of articles, we use acronyms as a last resort. But we defined these acronyms when they first appeared. We are very sorry about it.

Line 70. I think that you have to better introduce the degradation index to help the reader. I saw its explanation in section 2.3 but some details have to be introduced also in the introduction. Answer: Thank you for your suggestion. We have introduced the explanation of degradation index here.

The degradation index (DI) proposed by Dauwe et al. (1998, 1999) has been wildly used to assess the degradation state of organic materials (OM) in terrestrial, aquatic, and marine environment (Dauwe and Middelburg, 1998; Wang et al., 2018; Dauwe et al., 1999). This value is based on the molar percentage (Mol%) of the amino acid pool and higher DI values denote a more "fresh" state of protein matter.

Section 2.1. I think that you have to add some information about the type of filter used and the cleaning procedure of this filter. You have to add the reference but I think that you have to insert this information in the main manuscript. Answer: Thank you for your

suggestion. We have added more information about the type of filter used and the cleaning procedure of this filter.

Quartz fiber filters were used and filters were heated at 450°C for 10 h to remove any organics before sampling.

Section 3.2.3. This part is too short to be one section and I suggest to add this sentence to another section. Answer: Thank you for your suggestion. This part is incorporated into the section "the composition profiles of HAA in fine and coarse particles (Section 3.2.2)". Besides that, the structure of the entire result section has been adjusted.

Line 295. I don't understand why you use PC1 as coefficient. This principal component clearly distinguishes the fine and the coarse particles. I think that this point should be clarified in the manuscript. Answer: Thank you for your suggestion. The point that PC1 clearly distinguishes the fine and the coarse particles was clarified in the manuscript. For calculation of DI values for fine and coarse particles, the first principal component score from principal component analysis (PCA) was applied to our own data (including Ala, Gly, Val, Leu, Ile, Pro, Ser, Thr, Phe, Asp, Glu, Lys, His and Tyr, except GABA), following the method described by Dauwe et al. (1999). Fine and coarse particles were clearly distinguished by first principal component scores, suggesting that the first principal component score may also be designed as a degradation index of THAA in aerosols.

Section 3.3.4. I think that you have to define the meaning of DI values, also considering previous published results. You have to define the threshold when bacterial degradation occurred. I don't like so much this fragmentation of the section. This is only my opinion, but I think that this fragmentation produces to lose the thread. You have the sections with 4-5 lines. Answer: Thank you for your suggestion. The threshold of DI values was defined in this manuscript and the DI values of fine and coarse particles were compared with the previous published results.

In marine environment, high DI values (>0.5) indicate the better preservation of more

fresh organic matter from marine primary production (Jiang et al., 2014). On the contrary, low DI values (<0.5) indicate the presence of relatively degraded organic matter (Burdige, 2007; Wang et al., 2018). In this study, the DI values of fine particles were close to those of "fresh" material. For instance, source materials (e.g., plankton, bacteria and sediment trap material) and the precipitation in Uljin. On the contrary, the DI values of coarse particles were comparable to those of surface soil, POM in coastal sediments and DOM in coastal area and precipitation in Seoul, which were proved to be more degraded materials (Fig. 8).

In the conclusion, you affirm that "The difference in 15N values of Source-AA and THAA between coarse particles and fine particles were small," but one of the main aim of the manuscript is the follows: "15N values of Gly and THAA in fine and coarse particle were compared with those in main emission sources to identify the potential sources of fine and coarse particles.". So is the conclusion that 15N values are not good tracers to define the sources?

Answer: Sorry for our unclear description. "The difference in $\delta$15N values of Source-AA and THAA between coarse particles and fine particles." In this sentence, Source-AA is not the AAs released from potential emission sources. Here, Source-AA is defined as Gly, Val, Ser, Thr, Phe, Lys, His and Tyr, which is compared with Trophic-AA including Ala, Leu, Ile, Pro, Asp and Glu. This concept is proposed by McCarthy et al., 2007. They group individual AA based on the extensive metabolic diversity of microbes, coupled with their ability to salvage, resynthesize, or even alter synthetic pathways. The Trophic-AA group consists of aliphatic and acidic side chain AAs (Asp, Glu, Ala, Ile, Leu, and Val) as well as Pro, which has one N (excepting the associated forms Gln and Asn), which is obtained directly from Glu via transamination. The "scatting" $\delta$15N pattern of Trophic AA with total heterotrophic resynthesis is presumed. In contrast, the $\delta$15N pattern of Source-AA group remain relatively constant with total heterotrophic resynthesis includes Gly as well as most of the more chemically complex side chain AA (Ser, Thr, Phe, Tyr, and Lys). In this study, this "scattered" characteristic of $\delta$15N-AA

distribution in Trophic-AA group of coarse particles was observed while the difference in $\delta 15N$ values of Source-AA group between coarse particles and fine particles were small.

On the other hands, average $\delta 15N$ value for hydrolyzed Gly from the biomass burning, soil, and plant sources was +15.6 $\pm$ 4.3‰ +3.0 $\pm$ 4.4‰ and $-11.9\pm1.4$‰ respectively, and the mean $\delta 15NTHAA$ value was +15.8 $\pm$ 4.5‰ +5.5 $\pm$ 2.2‰ and $-0.0 \pm$ 1.8‰ respectively. $\delta 15N$ value for HAAs from the biomass burning sources are significantly higher than those observed from natural sources (plant and soil sources). Therefore, $\delta 15N$ value for hydrolyzed HAAs may be a good tracer to identify sources.

To avoid ambiguity, we specific the main emission sources in the aim." $\delta 15N$ values of Gly and THAA in fine and coarse particle were compared with those in main emission sources (biomass burning, soil and plant sources) to identify the potential sources of fine and coarse particles"

Specific comments Lines 42-43. I suggest rephrasing this part because I think that English form is not correct. For example you repeated "compound". Answer: This sentence was rephrased. Recently, an increasing number of researchers highlight the importance of amino acids (AAs) in the atmosphere because AA is considered to be one of the most important organic nitrogen compounds in atmosphere (Zhang et al., 2002; Matos et al., 2016).

Line 43. I suggest to insert this reference because it summarized very well the state of knowledge in the 2016: "Matos, João TV, Regina MBO Duarte, and Armando C. Duarte. "Challenges in the identification and characterization of free amino acids and proteinaceous compounds in atmospheric aerosols: a critical review." TrAC Trends in Analytical Chemistry 75 (2016): 97-107." Answer: Thank you for your suggestion. This reference was added. "Recently, an increasing number of researchers highlight the importance of amino acids (AAs) in the atmosphere because AA is considered to be one of the most important organic nitrogen compounds in atmosphere (Zhang et al.,

2002; Matos et al., 2016)."

Line 45. Here a reference is needed. Answer: Thank you for your suggestion. The reference was added. "Recently, an increasing number of researchers highlight the importance of amino acids (AAs) in the atmosphere because AA is considered to be one of the most important organic nitrogen compounds in atmosphere (Zhang et al., 2002; Matos et al., 2016). Moreover, AAs are bioavailable and can be directly utilized by plant and soil communities (Wedyan and Preston, 2008; Song et al., 2017). Its key role in atmosphere-biosphere nutrient cycling and global nitrogen cycle has aroused greatly concern (Samy et al., 2013; Zhang and Anastasio, 2003). Besides that, AAs and proteins are important constituents of allergenic bioaerosol (Miguel et al., 2009; Huffman et al., 2013). The distribution of AAs and proteins in different particle sizes will determine whether these compounds can reach the pulmonary alveoli and the allergy of aerosols (Di Filippo et al., 2014)."

Line 50. I suggest you this paper where the particle size distribution of free amino acids is investigated until nano dimension: "Barbaro, et al. "Characterization of the water soluble fraction in ultrafine, fine, and coarse atmospheric aerosol." Science of The Total Environment 658 (2019): 1423-1439.". Answer: Thank you for your suggestion. This reference was added and this sentence "However, detail information on the concentrations and mole composition profiles of AA distributed in different size particle is still limited." was deleted. "And the distribution of AAs associated with different particle sizes can help to trace the sources and transformation of atmospheric aerosols (Barbaro et al., 2019; Feltracco et al., 2019; Di Filippo et al., 2014)."

Line 67-69. I think that you should also add the investigation of Kuznetsova et al. "Kuznetsova, M., Lee, C., Aller, J., 2005. Characterization of the proteinaceous matter in marine aerosols. Mar. Chem. 96, 359e377. https://doi.org/10.1016/j.marchem.2005.03.007 Answer: Thank you for your suggestion. This reference was added. Unfortunately, bacterial degradation of atmospheric AAs is limited. For example, two studies on marine aerosols by Wedyan and Preston

(2008) and Kuznetsova et al. (2005), and one study on precipitation by Yan et al. (2015).

Line 114. Have you verified the recovery of amino acids from the cationic cation exchange column? Figure S1. Please add (F) and (C) in the caption after fine and coarse. Change "blue" with "green" because I saw green the coarse particles Answer: Yes, we verified the recovery of amino acids from the cationic cation exchange column. It has been published in our previous study. "Zhu, R.-g., et al. (2020). "Nitrogen isotopic composition of free Gly in aerosols at a forest site." Atmospheric Environment 222: 117179." See the table below. (F) and (C) in the caption after fine and coarse were added. "blue" was changed to"green". Thank you.

Lines 186-187 and in other sections of manuscript. Please consider to significant figures. For example, "2542.9±1820.1 pmol m-3" should be 2542±1820 or the best way is 3±2 nmol m-3. I found the same mistake in the % values. Answer: Sorry for our mistake. Significant figures of the concentration of HAA and % values were corrected in this manuscript.

Lines 421. Please consider that the combined amino acids were investigated also in the Arctic region, considering also the particle size distribution. Feltracco, et al. "Free and combined L-and D-amino acids in Arctic aerosol." Chemosphere 220 (2019): 412- Answer: Thank you for your suggestion. This reference was added. "Feltracco et al. (2019) demonstrated that free and combined amino acids in Arctic aerosol were mainly distributed in fine fraction, which could be affect by several sources, including biological primary production and biomass burning."

Lines 430-432. You have completely skipped the marine contribution. Several studies conducted by prof. Leck (Leck and Bigg, 2005a, 2005b; Bigg, 2007; Bigg and Leck, 2008) demonstrated the sea emission of PBAP. Combined amino acids is surely one of the main component of PBAP.

Answer: Sorry for our mistake. Indeed, as one of the main components of PBAP, AAs

are proved to be released by ocean (Leck and Bigg, 2005a, 2005b; Bigg, 2007; Bigg and Leck, 2008). Marine source may also contribute to atmospheric AAs for both fine and coarse particles observed here. However, the sampling sites are located in an inland city. Considering the 2-day back trajectory of during sampling periods (Fig. S5), we can observe that the aerosol collected flowed principally from the mainland and air mass from marine only accounted for 16%. Moreover, during the long transport, PABP may be removed by dry and wet deposition (Bespres et al., 2012). Therefore, in this study, compared to land origin, the contribution of marine source to aerosol AAs observed here may be relatively small. Unfortunately, we do not have $\delta$15N-HAA data for marine aerosols. Pooled $\delta$15NGly values from literature data, we found the $\delta$15NGly values in ocean high molecular weight dissolved organic matter, cyanobacteria and plankton ranged from -16.6‰ to +7.7‰ (McCarthy et al.,2007; Mcclelland and Montoya,2002; Chikaraishi et al., 2009; McClelland et al. 2003; Calleja et al., 2013), which were close to range of the natural source including plant (range: -13.2‰ to -9.7‰ and soil (range: -1.6‰ to +7.4‰ sources. Conclusively, the contribution from soil and plant sources mentioned in this study may possibly including a small amount of marine contribution.

Technical correction Line 23. Please remove one point from (p<0.0.1). Line 80. Please change as "particle sizes" Line 90. Change "was" with "were" Line 105. Please introduce the acronym HAA. Answer: Sorry for our mistake. All these technical mistakes were changed in this manuscript and acronym HAA was defined.

Please also note the supplement to this comment:
https://acp.copernicus.org/preprints/acp-2020-534/acp-2020-534-AC1-supplement.pdf
* * *
[Figure]

**Fig. 1.** 2-day (48 h) back trajectories illustrating the typical air mass flows to the sampling site

**Supplement:**

Supporting information

[revised manuscript text omitted]

Extraction Method: Principal Component Analysis.

Component Scores.

---

## Referee Comment (RC2) · Anonymous Referee #2 · 4 Oct 2020

The manuscript entitled "Measurement report: Amino acids in fine and coarse atmospheric aerosol: concentrations, compositions, sources and possible bacterial degradation state" by Zhu et al. shows the measurement of combined amino acids in fine (<2.5 $\mu$m) and coarse (>2.5 $\mu$m) aerosol particles. The isotopic ratios of 15N of individual amino acids were obtained and studied for source investigation. The degradation of microbiomes especially bacterial were discussed. The results are very interesting and should be suitable for publication in the journal. I also found that the manuscript was not very well prepared at the current stage and many parts could (and should) be

clarified before being considered for publication.

The title should be revised. Firstly, it is strange to mention "measurement report", isn't it? Is it really necessary? Secondly, "combined amino acids" should be clear. Otherwise, it may refer to free amino acids.

The application of isotopic ratios of stable nitrogen and degradation index would be very interesting for source investigation. In this work, the author may think about what they would really like to focus on and why they are important. The current manuscript contains data from observation and measurement but I feel it is a bit ambiguous on their conclusions. The Abstract could also be improved in order to present and show the main idea of this manuscript. By the way, isotopic ratio is a nice and promising tool for understanding the sources of combined amino acids but please note that the influence of atmospheric processes may affect the fractionation. The authors should also explain more of the connection between DI and bacterial degradation (especially in Section 4.2). At least for me, I could not really understand why? If I understood it correctly, amino acids were hydrolyzed and might present the composition of proteins. How could DI be used to estimate the bacterial degradation? The variation of DI may present the degree of aging but how to relate it to bacterial degradation?

The discussion in 4.1 leads to the conclusion that the sources of amino acids are somehow identical between fine and coarse particles. The question may come to the point of more degradation in coarse particle. If degradation is important for amino acids, some difference of composition profile or source contribution should be found between fine and coarse particles. The other possibility is that the support of $\delta 15N$ may not be sufficient in this case. PCA was used in this study. Why not putting more chemical species and organic tracers in the PCA analysis as many studies did? These amino acids were combined and should how could they contributed from different sources?

The discussion on the release of coarse "fresh" bioparticles at the onset of rainfall seems arbitrary and could be clarified with the support of precipitation data (how long

and how strong the rain happened). As is known, rainfall may promote the release of bioaerosols but it also depends on the frequency and intensity. In most cases, it occurs mainly in a much shorter time scale. The rainfall then could suppress the concentrations of bioaerosols in the air. The sampling was daily based in this study and it may not be very well to observe this variation in my personal opinion.

The part of Results contains many short sub-sections which is not very friendly for the readers. These sub-sections only repeat the Table and Figures, making it very hard to follow. Please re-arrange it. Why not put results and discussion together?

I would suggest the authors to select and keep some nice figures and move some to the supporting file. It may help to make the manuscript clear and concise.

Fig. 1 and Fig. 3: The information of fine/coarse particles is missing. I was confused by them and had to search which data belong to fine/coarse particles.

Minor mistakes/errors: Seems quite many minor mistakes/errors throughout the text. Please check through. For example: L24: "p<0.0.1" should be "p<0.1" L47: "allergy" should be "allergenicity"? L141: "PCi" should be deleted? I suggest to introduce more information of DI. L146: Rstandard should be described here. L159: Please check the formula. It seems wrong. L291: "Compared our calculating method with other works"? comparing the results or comparing the method?
* * *

---

## Referee Comment (RC3) · Anonymous Referee #3 · 13 Oct 2020

Comments

The research on amino acids in aerosols is an interesting topic, given their important roles in human health, air quality and even climate. In this study, Zhu and coauthors investigated the amino acids in fine and coarse aerosol particles collected from a city in China, with emphasis on their concentrations, compositions profile, as well as the potential controlling process. However, this current manuscript suffers from substantial weakness. Generally, the experimental design, including instrumentations, observation locations made it very hard to conduct in depth data interpretation and discussion.

Therefore, the current version of the manuscript is more like a data report, than a research article.

Specific comments: 1. The title is misleading. Is the knowledge from this study applicable universally? Actually, it seems only reflect the scenario at the sampling sites, i.e. Nangchang, China. So the information of the study area should be added in the title. 2. In the section 2.1 Sampling collection. What is the consideration to choose urban, town, town, suburban, airport and forest as reprehensive sites for the amino acids study? Especially, why airport was selected as one of the sampling sites? It seems agricultural site is more important than other areas for the amino acids. 3. Also Section 2.1, How many forest soil, paddy soil, road soil were collected and analyzed? 4. In the Section 2.2, I did not find the description of the pretreatment and chemical analysis for soil samples. Current contents are only about the analysis of aerosol samples. 5. In the results, Section 3.1.2, Line 191, It was found that the concentration level of THAA at airport is highest among the five different sites. But the reason was not appropriately explained later. 6. Line 198-204, here I understand that precipitation is an important factor to affect the amino acids in air, not only the concentration level, but also the composition pattern. However, actually there are many meteorological factors could change the amino acid in air. Why did you only consider the impact of precipitation? If you want to reveal the influence of precipitation on AA, actually the precipitation samples should also collected and analyzed simultaneously, along the aerosol sampling. The AA in precipitation samples could offer more information. 7. Line 208, here you mentioned the results of pine and straw in Figure2. Which kind of straw? Actually I did not find the relevant information in the sample collection section. 8. For the PCA analysis, there are different influencing factors (sources and degradation process ) for AA in the five sites. Is it appropriate to include the all data from different sites to conduct the PCA analysis? If the sample amount is enough, it seems more reasonable to just use the data for specific site, respectively, the reveal the difference between fine and coarse particles. 9. Line416-420, those sentences seem should be moved to Introduction part. 10. Generally, in the results and discussion, more description on the

novelty of this study is needed. What are the new findings from this study compared to what already known, and what the significance and implication of the new findings for others?

---

## Author Response (AR1)

**Dear Reviewer:**

**Thank you for your letter and comments concerning our manuscript acp-2020-534 entitled "Measurement report: Amino acids in fine and coarse atmospheric aerosol: concentrations, compositions, sources and possible bacterial degradation state". Those comments are all valuable and very helpful improving our paper. We accept your suggestions and revise the manuscript according to the points as follows. Particularly, we have re-arranged the results and discussion section.**

**Anonymous Referee #1**

**General comments**

**Abstract. In the abstract the use of acronyms is inappropriate as well as it is dissuaded to insert the references.**

Answer: Thank you for your suggestion. Since the samples analyzed in this paper were obtained in the same sampling campaign of Zhu et al. (2020), according to editor requirement, we referenced the Zhu et al. 2020 and clearly stated that the samples are the same in both papers but the data used in this manuscript were not analyzed in Zhu et al., (2020).

Besides that, we try our best to reduce the use of acronyms in the abstract. Gly, Ser, Phe and Lys have changed to the full name in the abstract. But we still preserved some acronyms, including amino acids (AAs), hydrolyzed amino acid (HAA), $\delta^{15}N$ values of total hydrolyzed amino acid ($\delta^{15}N_{THAA}$), degradation index (DI), and the variance within trophic AAs ($\sum V$). These acronyms occurred several times in the abstract (at least three times) and the full name of these acronyms are long. To reduce the length of the abstract and improve the readability of articles, we use acronyms as a last resort. But we defined these acronyms when they first appeared. We are very sorry about it.

**Line 70. I think that you have to better introduce the degradation index to help the reader. I saw its explanation in section 2.3 but some details have to be introduced also in the introduction.**

**Answer:** Thank you for your suggestion. We have introduced the explanation of degradation index here.

The degradation index (DI) proposed by Dauwe et al. (1998, 1999) has been wildly used to assess the degradation state of organic materials (OM) in terrestrial, aquatic, and marine environment (Dauwe and Middelburg, 1998; Wang et al., 2018; Dauwe et al., 1999). This value is based on the molar percentage (Mol%) of the amino acid pool and higher DI values denote a more "fresh" state of protein matter. Line 72-76.

**Section 2.1. I think that you have to add some information about the type of filter used and the cleaning procedure of this filter. You have to add the reference but I think that you have to insert this information in the main manuscript.**

Answer: Thank you for your suggestion. We have added more information about the type of filter used and the cleaning procedure of this filter. Line 100-101.

Quartz fiber filters were used and filters were heated at 450°C for 10 h to remove any organics before sampling.

**Section 3.2.3. This part is too short to be one section and I suggest to add this sentence to another section.**

Answer: Thank you for your suggestion. This part is incorporated into the section "Concentrations and mol% composition profile of HAA in size- segregated aerosol (Section 3.1)". Besides that, the structure of the entire result section has been adjusted. We have put the results and discussion part together as reviewer #2 suggested.

**Line 295. I don't understand why you use PC1 as coefficient. This principal component clearly distinguishes the fine and the coarse particles. I think that this point should be clarified in the manuscript.**

Answer: Thank you for your suggestion. The point that PC1 clearly distinguishes the fine and the coarse particles was clarified in the manuscript. Line 325-327 and Line 334-336.

For calculation of DI values for fine and coarse particles, the first principal component score from principal component analysis (PCA) was applied to our own data (including Ala, Gly, Val, Leu, Ile, Pro, Ser, Thr, Phe, Asp, Glu, Lys, His and Tyr, except GABA), following the method described by Dauwe et al. (1999). Fine and coarse particles were clearly distinguished by first principal component scores, suggesting that the first principal component score may also be designed as a degradation index of THAA in aerosols.

**Section 3.3.4. I think that you have to define the meaning of DI values, also considering previous published results. You have to define the threshold when bacterial degradation occurred.**
**I don't like so much this fragmentation of the section. This is only my opinion, but I think that this fragmentation produces to lose the thread. You have the sections with 4-5 lines.**

Answer: Thank you for your suggestion. In revised manuscript, the meaning of values, threshold of DI values was defined and the DI values of fine and coarse particles were compared with the previous published results.

"Protein as major components in all source organisms are sensitive to all stages of degradation (Cowie and Hedges 1992). Moreover, compared to the alteration of the degradation, the dissimilarity in amino acid composition of protein in the source organisms are minor (Dauwe and Middelburg 1998). Therefore, the degradation index (DI) is developed, which are based on protein amino acid composition and factor coefficients based on the first axis of the PCA analysis (equation 1). Since AAs concentrated in cell walls are preferential accumulated during decomposition, whereas amino acids that are concentrated in cell plasma tend to be depleted during degradation (Dauwe et al., 1999), the compositional changes of amino acids associated with degradation can be traced by the DI value. The higher DI values indicate the protein is relatively "fresh" (Yan et al., 2015) and changes tracked by DI are proposed to be driven in large part by enrichment of AAs concentrated in cell wall (Mccarthy et al., 2007).

The DI values of fine particles were close to those of "fresh" material. For instance, source materials (e.g., plankton, bacteria and sediment trap material). On the contrary, the DI values of coarse particles were comparable to those of surface soil, POM in coastal sediments and DOM in coastal area, which were proved to be more degraded materials (Fig. 8). In marine environment, high DI values ($>0.5$) indicate the better preservation of more fresh organic matter from marine primary production (Jiang et al., 2014). On the contrary, low DI values ($<0.5$) indicate the presence of relatively degraded organic matter (Burdige, 2007; Wang et al., 2018). In this study, the lower DI values observed in coarse particles, implying that AAs in coarse particles may undergo more degradation than fine particles. Our result is also comparable to that observed in precipitation at Uljin and Seoul (Yan et al., 2015). The DI values measured in coarse particles are closer to those observed in Seoul, where is believed to have more advanced bacterial degradation than Uljin, further supporting the degradation degree of amino acids in coarse particles is higher than that in fine particles.

Moreover, the section 3.3.4 in previous manuscript was re-arranged. All content concerning the meaning of DI and degradation state of AAs in size-segregated aerosol have been put into the section 3.4 "Different degradation state of AAs between fine and coarse aerosol particles".

**In the conclusion, you affirm that "The difference in 15N values of Source-AA and THAA between coarse particles and fine particles were small," but one of the main aim of the manuscript is the follows: "15N values of Gly and THAA in fine and coarse particle were compared with those in main emission sources to identify the potential sources of fine and coarse particles.". So is the conclusion that 15N values are not good tracers to define the sources?**

Answer: Sorry for our unclear description. "The difference in $\delta^{15}$N values of Source-AA and THAA between coarse particles and fine particles." In this sentence, Source-AA is not the AAs released from potential emission sources. Here, Source-AA is defined as Gly, Val, Ser, Thr, Phe, Lys, His and Tyr,

which is compared with Trophic-AA including Ala, Leu, Ile, Pro, Asp and Glu.

This concept is proposed by McCarthy et al., 2007. They group individual AA based on the extensive metabolic diversity of microbes, coupled with their ability to salvage, resynthesize, or even alter synthetic pathways. The Trophic-AA group consists of aliphatic and acidic side chain AAs (Asp, Glu, Ala, Ile, Leu, and Val) as well as Pro, which has one N (excepting the associated forms Gln and Asn), which is obtained directly from Glu via transamination. The ''scatting'' $\delta^{15}N$ pattern of Trophic AA with heterotrophic resynthesis is presumed.

In contrast, the $\delta^{15}N$ pattern of Source-AA group remain relatively constant with total heterotrophic resynthesis includes Gly as well as most of the more chemically complex side chain AA (Ser, Thr, Phe, Tyr, and Lys).

In this study, this ''scattered'' characteristic of $\delta^{15}N$-AA distribution in Trophic-AA group of coarse particles was observed while the difference in $\delta^{15}N$ values of Source-AA group between coarse particles and fine particles were small. This indicate that AAs in coarse particles have stronger bacterial degradation state than those in fine particles.

On the other hands, average $\delta^{15}N$ value for hydrolyzed Gly from the biomass burning, soil, and plant sources was $+15.6 \pm 4.3‰$, $+3.0 \pm 4.4‰$, and $-11.9 \pm 1.4‰$, respectively, and the mean $\delta^{15}N_{THAA}$ value was $+15.8 \pm 4.5‰$, $+5.5 \pm 2.2‰$, and $-0.0 \pm 1.8‰$, respectively. $\delta^{15}N$ value for HAAs from the biomass burning sources are significantly higher than those observed from natural sources (plant and soil sources). Therefore, $\delta^{15}N$ value for hydrolyzed HAAs may be a good tracer to identify sources.

To avoid ambiguity, we defined the main emission sources in the aim.'' $\delta^{15}N$ values of Gly and THAA in fine and coarse particle were compared with those in main emission sources (biomass burning, soil and plant sources) to identify the potential sources of fine and coarse particles'' Line 88-90.

**Specific comments**

**Lines 42-43. I suggest rephrasing this part because I think that English form is not correct. For example you repeated "compound".**

Answer: This sentence was rephrased. Recently, an increasing number of researchers highlight the importance of amino acids (AAs) in the atmosphere because AA is considered to be one of the most important organic nitrogen compounds in atmosphere (Zhang et al., 2002; Matos et al., 2016). Line 38-40.

**Line 43. I suggest to insert this reference because it summarized very well the state of knowledge in the 2016: "Matos, João TV, Regina MBO Duarte, and Armando C. Duarte. "Challenges in the identification and characterization of free amino acids and proteinaceous compounds in atmospheric aerosols: a critical review." TrAC Trends in Analytical Chemistry 75 (2016): 97-107."**

Answer: Thank you for your suggestion. This reference was added.

"Recently, an increasing number of researchers highlight the importance of amino acids (AAs) in the atmosphere because AA is considered to be one of the most important organic nitrogen compounds in atmosphere (Zhang et al., 2002; Matos et al., 2016)." Line 38-40.

**Line 45. Here a reference is needed.**

Answer: Thank you for your suggestion. The reference was added.

"Recently, an increasing number of researchers highlight the importance of amino acids (AAs) in the atmosphere because AA is considered to be one of the most important organic nitrogen compounds in atmosphere (Zhang et al., 2002; Matos et al., 2016). Moreover, AAs are bioavailable and can be directly utilized by plant and soil communities (Wedyan and Preston, 2008; Song et al., 2017). Its key role in atmosphere-biosphere nutrient cycling and global nitrogen cycle has aroused greatly concern (Samy et al., 2013; Zhang and Anastasio, 2003). Besides that, AAs and proteins are important constituents of allergenic bioaerosol (Miguel et al., 2009; Huffman et al., 2013). The distribution of AAs and proteins in different particle sizes will determine whether these compounds can reach the pulmonary alveoli and the allergy of aerosols (Di Filippo et al., 2014)." Line 38-48.

**Line 50. I suggest you this paper where the particle size distribution of free amino acids is investigated until nano dimension: "Barbaro, et al. "Characterization of the water soluble fraction in ultrafine, fine, and coarse atmospheric aerosol." Science of The Total Environment 658 (2019): 1423-1439.".**

Answer: Thank you for your suggestion. This reference was added and this sentence "However, detail information on the concentrations and mole composition profiles of AA distributed in different size particle is still limited." was deleted. It was changed to "And the distribution of AAs associated with different particle sizes can help to trace the sources and transformation of atmospheric aerosols (Barbaro et al., 2019; Feltracco et al., 2019; Di Filippo et al., 2014)." Line 46-48.

**Line 67-69. I think that you should also add the investigation of Kuznetsova et al. "Kuznetsova, M., Lee, C., Aller, J., 2005. Characterization of the proteinaceous matter in marine aerosols. Mar. Chem. 96, 359e377. https://doi.org/10.1016/j.marchem.2005.03.007**

Answer: Thank you for your suggestion. This reference was added.

Unfortunately, bacterial degradation of atmospheric AAs is limited. For example, two studies on marine aerosols by Wedyan and Preston (2008) and Kuznetsova et al. (2005), and one study on precipitation by Yan et al. (2015). Line 70-72.

**Line 114. Have you verified the recovery of amino acids from the cationic cation exchange column?**
**Figure S1. Please add (F) and (C) in the caption after fine and coarse. Change "blue" with "green" because I saw green the coarse particles**

**Answer: Yes, we verified the recovery of amino acids from the cationic cation exchange column. It has been published in our previous study. "**Zhu, R.-g., et al. (2020). "Nitrogen isotopic composition of free Gly in aerosols at a forest site." Atmospheric Environment **222**: 117179.**" See the table below.**

**(F) and (C) in the caption after fine and coarse were added in Figure S1. "blue" was changed to"green". Thank you.**

Analytical characteristics for amino acid derivatives using GC–MS (full scan) method. Correlation coefficients obtained from linear regression analysis of calibration curves. Instrumental limit of detection (LOD) based on a signal-to-noise ratio of 3. Instrumental limit of quantification (LOQ) based on a signal-to-noise ratio of 10. EMDL was the corresponding effective limit in the aerosol samples.

| Amino acids | Retention time | FAA % recovery | CAA % recovery | Correlation coefficient ($r^2$) | LOD (pmol) | LOQ (pmol) | EMDL (pmol m-3) |
|---|---|---|---|---|---|---|---|
| Alanine (Ala) | 22.1 | 103±4 | 94±3 | 0.9928 | 0.1 | 0.3 | 0.1 |
| Glycine (Gly) | 22.8 | **97±5** | **103±25** | 0.9948 | 0.1 | 0.5 | 0.1 |
| Valine (Val) | 26.4 | 98±3 | 97±3 | 0.9936 | 0.1 | 0.3 | 0.1 |
| Leucine (Leu) | 27.8 | 96±1 | 95±6 | 0.9917 | 0.1 | 0.3 | 0.1 |
| Isoleucine (Ile) | 28.9 | 94±1 | 93±1 | 0.9930 | 0.1 | 0.3 | 0.1 |
| γ-Aminobutyric acid (Gaba) | 29.8 | 95±3 | 92±6 | 0.9955 | 0.2 | 0.7 | 0.2 |
| Proline (Pro) | 30.2 | 101±10 | 74±2 | 0.9975 | 0.7 | 2.3 | 0.7 |
| Methionine (Met) | 36.5 | 99±5 | 91±5 | 0.9946 | 0.1 | 0.3 | 0.1 |
| Serine (Ser) | 37.1 | 101±5 | 84±2 | 0.9883 | 0.1 | 0.4 | 0.1 |
| Threonine (Thr) | 37.8 | 83±11 | 91±5 | 0.9891 | 0.1 | 0.3 | 0.1 |
| Phenylalanine (Phe) | 39.0 | 82±1 | 84±2 | 0.9921 | 0.1 | 0.2 | 0.1 |
| Aspartic acid (Asp) | 40.0 | 96±2 | 86±18 | 0.9914 | 0.1 | 0.4 | 0.1 |
| Glutamic acid (Glu) | 41.7 | 96±1 | 112±20 | 0.9868 | 0.9 | 3.1 | 1.0 |
| Asparagine (Asn) | 42.1 | 89±5 | NA | 0.9978 | 1.7 | 5.8 | 1.8 |
| Lysine (Lys) | 43.2 | 93±7 | 67±5 | 0.9865 | 0.3 | 1.0 | 0.3 |
| Glutamine (Gln) | 43.8 | 95±7 | NA | 0.9959 | 0.8 | 2.8 | 0.9 |

| | | | | | | | |
|---|---|---|---|---|---|---|---|
| **Arginine (Arg)** | 44.8 | 86±9 | 60±6 | 0.9969 | 0.9 | 2.9 | 0.9 |
| **Histidine (His)** | 46.6 | 95±15 | 92±5 | 0.9944 | 2.8 | 9.2 | 2.9 |
| **Tyrosine (Tyr)** | 47.4 | 79±1 | 52±2 | 0.9954 | 0.3 | 0.8 | 0.3 |
| **Tryptophan (Trp)** | 48.4 | 80±4 | NA | 0.9917 | 14.4 | 48.0 | 15.0 |

**Lines 186-187 and in other sections of manuscript. Please consider to significant figures. For example, "2542.9±1820.1 pmol m-3" should be 2542±1820 or the best way is 3±2 nmol m-3. I found the same mistake in the % values.**

Answer: Sorry for our mistake. Significant figures of the concentration of HAA and % values were corrected in revised manuscript.

**Lines 421. Please consider that the combined amino acids were investigated also in the Arctic region, considering also the particle size distribution. Feltracco, et al. "Free and combined L- and D-amino acids in Arctic aerosol." Chemosphere 220 (2019): 412-**

Answer: Thank you for your suggestion. This reference was added.

"Feltracco et al. (2019) demonstrated that free and combined amino acids in Arctic aerosol were mainly distributed in fine fraction, which could be affect by several sources, including biological primary production and biomass burning." Line 23-241.

**Lines 430-432. You have completely skipped the marine contribution. Several studies conducted by prof. Leck (Leck and Bigg, 2005a, 2005b; Bigg, 2007; Bigg and Leck, 2008) demonstrated the sea emission of PBAP. Combined amino acids is surely one of the main component of PBAP.**

Answer: Sorry for our mistake. Indeed, as one of the main components of PBAP, AAs are proved to be released by ocean (Leck and Bigg, 2005a, 2005b; Bigg, 2007; Bigg and Leck, 2008). Marine source may also contribute to atmospheric AAs for both fine and coarse particles observed here. However, the sampling sites are located in an inland city. Considering the 2-day back trajectory of during sampling periods (Fig. S2), we can observe that the aerosol collected flowed principally from the mainland and air mass from marine only accounted for 16%. Moreover, during the long transport, PABP may be removed by dry and wet deposition (Bespres et al., 2012). Therefore, in this study, compared to land origin, the contribution of marine source to aerosol AAs observed here may be relatively small. Unfortunately, we do not have $\delta^{15}$N-HAA data for marine aerosols. Pooled $\delta^{15}N_{Gly}$ values from literature data, we found the $\delta^{15}N_{Gly}$ values in ocean high molecular weight dissolved organic matter, cyanobacteria and plankton ranged from -16.6‰ to +7.7‰ (McCarthy et al.,2007; Mcclelland and Montoya,2002; Chikaraishi et al., 2009; McClelland et al. 2003; Calleja et al., 2013), which were close to range of the natural source including plant (range: -13.2‰ to -9.7‰) and soil (range: -1.6‰ to +7.4‰) sources. Conclusively, the contribution from soil and plant sources mentioned in this study may possibly including a small amount of marine contribution. Line 275-289.

2-day (24 h) back trajectories was added in the supporting information (Fig. S2).

[Figure]

Figure S2. The 2-day (48 h) back trajectories illustrating the typical air mass flows to the sampling site (28.85°N, 115.91°E) during the sampling periods. The map comes from the MeteoInfoMap (version 1.4.9R2) software (Chinese Academy of Meteorological Sciences, China).

**Technical correction Line 23. Please remove one point from (p<0.0.1). Line 80. Please change as "particle sizes" Line 90. Change "was" with "were" Line 105. Please introduce the acronym HAA.**

Answer: Sorry for our mistake. All these technical mistakes were changed in this manuscript and acronym HAA was defined.

**Anonymous Referee #2**
**The title should be revised. Firstly, it is strange to mention "measurement report",**
**isn't it? Is it really necessary? Secondly, "combined amino acids" should be clear.**
**Otherwise, it may refer to free amino acids.**

Answer: Thank you for your suggestion. According to the suggestion of the editor, the type of this article was changed to the measurement report rather than research article. Therefore, "measurement report" should be added in the title according to the submission requirements. Furthermore, "Hydrolyzed amino acids" was added in the title in order to differential "free amino acids".

**The application of isotopic ratios of stable nitrogen and degradation index would be very interesting for source investigation. In this work, the author may think about what they would really like to focus on and why they are important. The current manuscript contains data from observation and measurement but I feel it is a bit ambiguous on their conclusions. The Abstract could also be improved in order to present and show the main idea of this manuscript.**

Answer: Thank you for your suggestion. The abstract was improved. In this work, we focus on the distribution, sources and possible bacterial degradation state of HAAs in size-segregated aerosol (>2.5μm and PM2.5). This work is important for uncovering the origin, transformation and fate of HAAs in the atmosphere. In revised abstract, we focus on the main idea of this manuscript and the significance of the work.

This size distribution of AAs can help understand its transformation and fate in the atmosphere. However, detailed information on this topic is limited to a few studies and very variable results for the size-segregated concentrations and mole composition of atmospheric combined AAs have been observed in previous studies. The factors controlling this large difference between fine and coarse particle are still unclear. Thus, verification of the different types, concentrations, origin and atmospheric processes of AAs distribution along the different air particle sizes is important and meaningful.

This study presents the first isotopic evidence that the sources of AAs for fine and coarse aerosol particles may be similar, all of which were influenced by biomass burning, soil, and plant sources.

It is still unknown that whether bacterial degradation play a role in the levels and compositions of AAs in different particle sizes. This is the first report of using degradation marker (DI) to investigate the degradation state of aerosol particles. Fine particles had significantly higher DI values than that of coarse particles ($p<0.05$), suggesting the degradation degree of amino acids in coarse particles is higher than that in fine particles.

Combining new compound-specific nitrogen isotope tool ($\delta^{15}$N-HAA) and effective bacterial heterotrophy indicator ($\sum V$), this study firstly provide evidence that the stronger degradation state the found in coarse particles are coupled with more bacterial heterotrophic resynthesis occurred in coarse particles.

In conclusion, the difference in the THAA concentration and mol% composition distribution between fine and coarse particles may be closely related to the stronger bacterial degradation of AAs occurred in coarse particles than those in fine particles.

**By the way, isotopic ratio is a nice and promising tool for understanding the sources of combined amino acids but please note that the influence of atmospheric processes may affect the fractionation.**

Answer: Thank you for your suggestion. It is true that atmospheric processes may affect the fractionation. We focus on whether AAs in fine or coarse undergo more atmospheric processes. In order to assess the fractionation of the atmospheric processes (oxidation, nitration and oligomerization of AA), we compared $\delta^{15}N$ values of AA in both fine and coarse particles. If AA in fine or coarse particles undergo particularly more photochemical transformation than the other, nitrogen isotopic fractionation during atmospheric processes could lead to the difference in $\delta^{15}N$ values of AA between fine and coarse particles. We found the difference in $\delta^{15}N$ values of Source-AA (Gly, Ser, Phe and Lys) and total hydrolysable amino acids ($\delta^{15}N_{THAA}$) between coarse particles and fine particles was relatively small (Fig. 3 and Fig. 4c). The average offset of $\delta^{15}N_{THAA}$ value between fine and coarse particles was lower than 1.5‰ (Fig. 4a). These results appear to contrast with what one might expect for protein AA in either sizes particles undergo particularly more photochemical transformation than the other.

**The authors should also explain more of the connection between DI and bacterial degradation (especially in Section 4.2). At least for me, I could not really understand why? If I understood it correctly, amino acids were hydrolyzed and might present the composition of proteins. How could DI be used to estimate the bacterial degradation? The variation of DI may present the degree of aging but how to relate it to bacterial degradation?**

Answer: Thank you for your suggestion. The relationship between DI and bacterial degradation were not clearly stated. Amino acids have been used to estimate the relative degradation state of the organic matter. Proteins are ubiquitous components of all source organisms and degradation mixtures (Cowie and Hedges 1992). Although there is some dissimilarity in amino acid composition of the ultimate source organisms (e.g., diatoms, coccolithophorids, and bacteria) (Cowie and Hedges 1992), these differences are minor compared to the alteration of the spectra upon degradation (Dauwe and Middelburg 1998). This value is based on the molar percentage (Mol%) of the amino acid pool and higher DI values denote a more "fresh" state of protein matter. If only use DI as indicator, the variation of DI may only present the extent of the degradation. The mistake in our previous manuscript has been corrected. Section 3.4.

However, the negative correlation of the DI with the concentration of free γ-aminobutyric acid (GABA) and its mole percentage are depicted in Figure S7. Since bacteria are known to produce free GABA from their protein precursors (Cowie and Hedges 1994; Koolman and Roehme, 2005), the concentrations and mole percentage of free GABA may tend to increase during the biodegradation process. Therefore, negative relationship between the DI values and GABA in aerosol suggested that the degradation of atmospheric protein is probably induced by bacteria. Dauwe et al. (1999) have also reported that the negative correlation of the DI with the mole percentage of the GABA and β-alanine (BALA) in marine particulate matter samples and they attributed the correlation of the DI with the variation of GABA mole percentage to the stimulation of degradation by the activity of microorganism. Furthermore, the ''scattered'' characteristic of $\delta^{15}$N-AA distribution in Tr-AA group of coarse particles and significant higher values of $\sum V$ were measured in coarse particles compared to fine particles (p<0.05). These further supporting the higher degradation degree of amino acids in coarse particles than that in fine particles are closely related to more bacterial heterotrophic resynthesis occurred in coarse particles.

**The discussion in 4.1 leads to the conclusion that the sources of amino acids are somehow identical between fine and coarse particles. The question may come to the point of more degradation in coarse particle. If degradation is important for amino acids, some difference of composition profile or source contribution should be found between fine and coarse particles. The other possibility is that the support of _15N may not be sufficient in this case. PCA was used in this study. Why not putting more chemical species and organic tracers in the PCA analysis as many studies did? These amino acids were combined and should how could they contributed from different sources?**

Answer: Sorry for our unclear description and thank you for your suggestion. Your assumption is right. In this work, the composition profiles of HAA in fine particles are quite different from those in coarse particles (Fig. 2). Both source contribution and the degradation process may cause the difference of composition profile of AAs. In this work, we measured the nitrogen isotopic compositions of hydrolyzed AAs released from main emission sources in the study areas, including biomass burning, soil and local plants (Fig. 3). The mean $\delta^{15}$N$_{THAA}$ value from the biomass burning, soil, and plant sources was +15.8 ± 4.5‰, +5.5 ± 2.2‰, and −0.0 ± 1.8‰, respectively. The differences between $\delta^{15}$N value of Gly among biomass burning source and plant sources up to 15.8‰. If either particle is more affected by biomass burning sources, increased particle $\delta^{15}$N$_{THAA}$ value would be observed and larger offset of $\delta^{15}$N$_{THAA}$ between fine and coarse would be expected, vice versa. However, the average offset of $\delta^{15}$N$_{THAA}$ value between fine and coarse particles was lower than 1.5 ± 1.7‰ in each sampling site, demonstrating the sources of AAs for fine and coarse aerosol particles may be similar. It is therefore that degradation processes cause the difference of

composition profile of AAs between fine and coarse particles.

Our purpose of using the PCA analyses is explore the influence of degradation process on the percentage profile of AAs rather than tracing source of AAs. In order to calculate the degradation index (DI), PCA used in this study and the first principal component score from principal component analysis (PCA) was applied to our own data (including Ala, Gly, Val, Leu, Ile, Pro, Ser, Thr, Phe, Asp, Glu, Lys, His and Tyr, except GABA), following the method described by Dauwe et al. (1999).

$$DI = \sum_i (\frac{Var_i - Avg_i}{SD_i}) \times PC1_i$$

Where $PC1_i$ is the loading of the amino acid $i$ obtained from PC1. If we using more chemical species and organic tracers in the PCA analysis as many studies did, we cannot obtain this key parameter (PC1 loading of each amino acids).

Combined amino acids in aerosol also could come from different sources. There are some reports suggested primary biological aerosol particles, fugitive dust, biomass burning, and agricultural or human activities (Kang et al., 2012; Matos et al., 2016). Therefore, we investigated the main emission sources in the study areas, including biomass burning (straw burning), soil (road soil, paddy soil and forest soil) and local plants (needles of pine and leaves of camphor). Line 49-53.

**The discussion on the release of coarse "fresh" bioparticles at the onset of rainfall seems arbitrary and could be clarified with the support of precipitation data (how longer and how strong the rain happened). As is known, rainfall may promote the release of bioaerosols but it also depends on the frequency and intensity. In most cases, it occurs mainly in a much shorter time scale. The rainfall then could suppress the concentrations of bioaerosols in the air. The sampling was daily based in this study and it may not be very well to observe this variation in my personal opinion.**

Answer: Thank you for your suggestion. We are sorry for our arbitrary. The frequency and intensity of rainfall during the sampling day were added in the supplementary materials. By comparing the data of the daily precipitation amount and the temporal variations of the concentration and mol% composition of HAA for coarse particle. We found the higher concentration of THAA was occurred in April 30, May 5, May 6 and May 13 when daily precipitation amount was higher and duration of rainfall was longer. Simultaneously, the mol% composition of HAA on those days were significantly different from that observed on dry days. These imply that rainfall may promote the release of bioaerosols but it depends on the rainfall amounts and intensity as your suggested. The stronger and longer rainfall events may promote more "fresh" protein matters in coarse aerosol. The discussion and conclusion have been changed in the manuscript. Line 511-527.

Table S4. Daily precipitation amount(mm) and hourly rainfall(mm) of sampling day.

| Date | Daily precipitation amount(mm) | Rainfall duration(hour) | Hourly rainfall(mm) |
|---|---|---|---|
| April 30 | 27.2 | 15 | 1.8 |
| May 1 | - | - | - |
| May 2 | - | - | - |
| May 3 | - | - | - |
| May 4 | - | - | - |
| May 5 | 1 | 3 | 0.3 |
| May 6 | 1 | 6 | 0.2 |
| May 7 | 0.6 | 9 | 0.1 |
| May 8 | 0.2 | 6 | 0.03 |
| May 9 | - | - | - |
| May 10 | - | - | - |
| May 11 | - | - | - |
| May 12 | 0.1 | 3 | 0.03 |
| May 13 | 31.4 | 12 | 2.6 |

-represent no precipitation.

**The part of Results contains many short sub-sections which is not very friendly for the readers. These sub-sections only repeat the Table and Figures, making it very hard to follow. Please re-arrange it. Why not put results and discussion together?**

Answer: Thank you for your suggestion. We have rearranged the part of results and put the results and discussion together.

**I would suggest the authors to select and keep some nice figures and move some to the supporting file. It may help to make the manuscript clear and concise.**

Answer: Thank you for your suggestion. The figures have been streamlined and move some to the supporting file. The total number of figures in revised manuscript is 9.

**Fig. 1 and Fig. 3: The information of fine/coarse particles is missing. I was confused by them and had to search which data belong to fine/coarse particles.**

Answer: Sorry for our negligence. Fine and coarse particles were added to Fig. 1 and Fig. 3. Thank you.

**Minor mistakes/errors: Seems quite many minor mistakes/errors throughout the text. Please check through. For example: L24: "p<0.0.1" should be "p<0.1" L47: "allergy" should be "allergenicity"? L141: "PCi" should be deleted? I suggest to introduce more information of DI. L146: Rstandard should be described here. L159: Please check the formula. It seems**

**wrong. L291: "Compared our calculating method with other works"? comparing the results or comparing the method?**

Answer: Sorry for our mistake.

L24: "p<0.0.1" was changed to "p<0.01".

L47: "allergy" was changed to "allergenicity".

L141: "PCi" was deleted and more information of DI was added. Line 156-159.

L146: Rstandard should be described here.

Rstandard is atmospheric $N_2$. Moreover, a derivatized mixture of 20 amino acid standards (Ala, Gaba, Arg, Asn, Asp, Gln, Glu, Gly, His, Ile, Leu, Lys, Met, Phe, Pro, Ser, Thr, Trp, Tyr, and Val) and several international amino acid standards (Ala, Gly3, Gly4, Phe, USGS40, USGS41a, and Val) with known $\delta^{15}N$ values (−26.35 to +47.55‰) was prepared to assess the isotope measurement reproducibility and normalize the $\delta^{15}N$ values of the amino acids in the samples (Zhu et al., 2018). Line 165-173.

**L159: Please check the formula.**

Sorry for our mistake. The formula was changed.

$$\delta^{15}N_{THAA} = \sum(\delta^{15}N_{HAA} \cdot mol\%HAA)$$

Where mol%HAA is the mole contribution of each HAA and $\delta^{15}N_{HAA}$ is the $\delta^{15}N$ value of individual HAA.

**It seems wrong. L291: "Compared our calculating method with other works"? comparing the results or comparing the method?**

It is wrong. It is comparing our results with other works. We have modified.

**Anonymous Referee #3**

**Specific comments: 1. The title is misleading. Is the knowledge from this study applicable universally? Actually, it seems only reflect the scenario at the sampling sites, i.e. Nangchang, China. So the information of the study area should be added in the title.**

Answer: Thank you for your suggestion. The information of the study area (Nanchang, China) has been added in the title. In this work, 5 sampling sites with different potential emission sources of atmospheric AAs were investigated in this study (See table below for details). The purpose of this work is using compound-specific $\delta^{15}N$ patterns of hydrolyzed amino acid (HAA), $\delta^{15}N$ values of total hydrolyzed amino acid ($\delta^{15}N_{THAA}$), degradation index (DI), and the variance within trophic AAs ($\sum V$) as markers to examine the sources and processing history of fine and coarse aerosol particles (>2.5μm and PM2.5). Our results demonstrated that the large difference in the concentration and mole percentage between fine and coarse particles might be closely related to the biologically relevant degradation processes. There is no particular situation in our sampling area (Nanchang, China). The results obtained in this work reflect a general law.

Table S1. The Characteristics and potential emission sources of atmospheric amino acids at 5 sampling area.

| Sampling sites | Characteristics and potential emission sources of atmospheric amino acids at sampling area |
|---|---|
| Urban | an area with dense population and human activities, industrial emissions, biomass burning and road dust |
| Town | local people cook and heat using straw, charcoal and wood, an area is more influenced by biomass burning |
| Suburban | a convergence area between city and rural, influenced by mixture sources, including biomass burning, agricultural activities and natural sources |
| Agricultural area | open area, surrounded by paddy fields, affected by agricultural activities |
| Forest | more affected by natural source, including viruses, algae, fungi, bacteria, protozoa, spores and pollen, fragments of plants and insects |

**2. In the section 2.1 Sampling collection. What is the consideration to choose urban, town, town, suburban, airport and forest as reprehensive sites for the amino acids study? Especially, why airport was selected as one of the sampling sites? It seems agricultural site is more important than other areas for the amino acids.**

Answer: Sorry for our unclear description. The concentration and composition of amino acids in the aerosols vary widely at different environment scenarios (Barbaro et al., 2011; Mace et al., 2003; Matsumoto et al., 2017; Mandalakis et al., 2011; Wang et al., 2019; Samy et al., 2011; Samy et al., 2013), and polar regions (Scalabrin et al., 2012; Barbaro et al., 2015).). This variability is highly dependent on the sources of AAs and meteorological conditions. Generally, atmospheric AAs is from primary biological aerosol particles (e.g., pollen, bacteria, fungi, spores, and fragments of living things), biomass burning, and agricultural activities (Kang et al., 2012; Matos et al., 2016). Therefore, the selection of sampling sites is based on the different emission sources of atmospheric amino acids in different characteristic areas of the city. As described in Table S1, from urban, town, suburban, airport and forest, the main emission sources of atmospheric AAs may change from the biomass burning sources to agricultural and natural sources. The airport is an open area, which is far away from the Nanchang city and surrounded by paddy fields. So, we suppose this sampling site is more affected by agricultural activities and natural sources. The airport is just a name of this sampling site, the real characteristic of this site was ignored. So, in revised manuscript, the name of the airport has been changed to agricultural area. This is also supported by the $\delta^{15}N_{Gly}$ and $\delta^{15}N_{THAA}$

signature. Compared with the town and urban areas, airport area (agricultural area) has relatively low $\delta^{15}N_{Gly}$ and $\delta^{15}N_{THAA}$ value in both fine and coarse particles ($P < 0.05$), which supporting that the atmospheric AA of fine and coarse particles in this site was more affected by agricultural and natural sources.

Thank you for your suggestion. The description of the characteristics and potential emission sources of atmospheric amino acids at 5 sampling area were added in supplement materials (Table S1).

**3. Also Section 2.1, How many forest soil, paddy soil, road soil were collected and analyzed?**

Answer: Sorry for our unclear description. For each type of soil samples, triplicate representative soil samples (approximately 100 g) were collected. Line 108-109.

**4. In the Section 2.2, I did not find the description of the pretreatment and chemical analysis for soil samples. Current contents are only about the analysis of aerosol samples.**

Answer: Sorry for our mistake. The description of the pretreatment and chemical analysis for soil and plant samples has been added in section 2.2. For plant and soil samples, approximately 30-40mg of plant or 500-600mg of soil powder were ground separately in liquid nitrogen into fine powders using a mortar and pestle. Then, well ground and homogenized soil and plant power were hydrolyzed in the same way as the aerosol samples. Line128-130.

**5. In the results, Section 3.1.2, Line 191, It was found that the concentration level of THAA at airport is highest among the five different sites. But the reason was not appropriately explained later.**

**Answer:** Sorry for our mistake. Indeed, the concentration level of THAA at airport is highest among the five different sites. The presence of amino acids in the atmosphere has been assessed in different environmental scenarios, e.g., urban (Barbaro et al., 2011; Wang et al., 2019), suburban (Samy et al., 2013), rural (Mace et al., 2003; Samy et al., 2011), marine (Mace et al., 2003; Matsumoto et al., 2017; Mandalakis et al., 2011), and polar regions (Scalabrin et al., 2012; Barbaro et al., 2015). The concentration of atmospheric amino acids in different environmental scenarios vary widely. This variability is highly dependent on the emission sources of AAs to the atmosphere and meteorological conditions. Since the distance between 5 sampling sites is less than 30 kilometers, it is therefore unlikely that meteorological conditions exert a major influence on the variability between sampling sites. So, the difference between the concentration level of THAA among the five different sites could be attributed to the emission sources of atmospheric AAs. The airport is an open area, which is far away from the city center and surrounded by paddy fields. Thus, it would be reasonable to deduce that in spring, enhanced agricultural activities and natural source emission (e.g., pollen grain) may lead to an increase in the concentration level of atmospheric AAs at the airport location. Line 222-225.

**6. Line 198-204, here I understand that precipitation is an important factor to affect the amino acids in air, not only the concentration level, but also the composition pattern. However, actually there are many meteorological factors could change the amino acid in air. Why did you only consider the impact of precipitation? If you want to reveal the influence of precipitation on AA, actually the precipitation samples should also collected and analyzed simultaneously, along the aerosol sampling. The AA in precipitation samples could offer more information.**

Answer: Thank you for your suggestion. We did not consider other meteorological conditions because except for precipitation, other meteorological conditions are not the main factors determining the concentration and percentage of amino acids in the atmosphere. The complexity of the results obtained for the potential influence of the meteorological conditions on the atmospheric levels of HAA at different environment scenarios (Mandalakis et al., 2011; Barbaro et al., 2015; Samy et al., 2013). With limited availability of such data, previous studies do not provide any definitive conclusion regarding the influence of other meteorological conditions on the atmospheric levels of HAA.

On the contrary, a tight relationship between atmospheric bioaerosols and precipitation has been found by previous studies (Huffman et al., 2013; Yue et al., 2016). Since amino acids predominantly exist as zwitterions (i.e., with protonated amine groups and deprotonated carboxylic acid groups) in aerosol, they will be found exclusively in condensed phases and their gas phase reactions do not need to be considered (Anastasio and McGregor, 2000). It is expected that the concentrations of individual AAs in aerosol are control by 2 mechanisms: the first one is the precipitation scavenging (Gorzelska and Galloway, 1990). This mechanism is supported by our recently work (Xu et al., 2020). We found that particulate AAs in precipitation are closely associated with aerosol particles and cloud condensation nuclei as a result of rainout and wash out effects (Xu et al., 2020). The second mechanism is that droplets splashing on the porous medium can deliver fresh biological aerosols in porous medium to the aerosol (Joung and Buie, 2015; Huffman et al., 2013; Yue et al., 2016). The first mechanism would certainly decrease the concentration of AAs in the aerosol, whereas the second mechanism may enhance the concentration of AAs. It is interesting to note that the average concentrations of THAA in coarse particles displayed no significant changes during rain events (p>0.05). For coarse particles, the average concentrations of THAA on rainy and dry days was 660.3±947.4 pmol m-3 and 212.2± 266.8 pmol m-3, respectively (Fig. 1 and Fig. S1). Owing to the high scavenging ratio of AAs in aerosol, the concentrations of individual AAs in aerosol were assumed to decrease during rainfall events. On the contrary, the concentrations of THAA in coarse particles displayed no significant changes during rain events, indicating particle emission mechanism (the second mechanism) were stronger than the precipitation scavenging mechanism (the first mechanism).

Moreover, the sources of AAs in coarse particles released by the particle emission mechanism are derived from porous medium rather than rainfall. These sources are primarily biological origin, including bacteria or fungal spores released from surrounding vegetation surfaces through mechanical agitation, spores ejected by fungi, lichens and other cryptogamic covers growing on soil, rock and vegetation (Elbert et al., 2007, 2012; Huffman et al., 2013). The principle of particle emission mechanism is mechanical agitation on porous medium and not closely related to the chemical composition of rainfall. Section 3.6.

**7. Line 208, here you mentioned the results of pine and straw in Figure2. Which kind of straw? Actually I did not find the relevant information in the sample collection section.**

Answer: Sorry for our mistake. The information of the pine and camphor have been added in the section 2.1. Masson pine (*Pinus massoniana* (Lamb.)) and camphor (*Cinnamomum Camphora*) tree as a common vegetation in the study area (115.8°E, 28.8°N) were collected during May 2019. Approximately 4-6 g of pine needles or camphor leaves were collected from the outer branches in the east, south, north, and west directions (about 10 m above the ground). We collected 5-6 representative samples for each kind of leaves.

All fresh samples were placed in plastic bags, labeled and stored in a chilled box immediately. In the laboratory, all plant and soil samples were freeze-dried. Then, freeze-dried samples were stored at -80°C until further use. Line 106-116.

**8. For the PCA analysis, there are different influencing factors (sources and degradation process ) for AA in the five sites. Is it appropriate to include the all data from different sites to conduct the PCA analysis? If the sample amount is enough, it seems more reasonable to just use the data for specific site, respectively, the reveal the difference between fine and coarse particles.**

Answer: Thank you for your suggestion. Proteins as major components in all source organisms are sensitive to all stages of degradation (Cowie and Hedges 1992). Moreover, Dauwe and Middelburg (1998) proved that the dissimilarity in amino acid composition of protein in the source organisms are minor, compared to the alteration of the degradation. It is therefore that the dissimilarity in the composition of protein between fine and coarse particles is caused by the degradation processes of protein rather than emission sources. So, in order to compare the changes of AAs percentage caused by degradation processes between fine and coarse particles, we use the all data to perform PCA analysis.

In this study, although the emission sources of atmospheric HAA were different among 5 sampling sites, the differences in DI values were not significant among 5 sampling sites for both fine and coarse particles ($p > 0.05$) (Fig. S6). For fine particles, the average DI values in airport, urban, forest, town and suburban location was 0.6±0.4, 0.5±0.5, 0.7±0.3, 0.6±0.3 and 0.7±0.2, respectively. For

coarse particles, the mean DI values in airport, urban, forest, town and suburban location was -0.5±0.9, -1.0±1.1, -0.8±1.1, -0.3±1.1 and -0.5±1.1, respectively. This result suggested that the degradation process of amino acids in the atmosphere is less affected by their emission sources. We are sorry for the lack of introduction to the DI concept in the previous manuscript.

**9. Line416-420, those sentences seem should be moved to Introduction part.**

Answer: Thank you for your suggestion. These sentences have been moved to the introduction part. Line 49-53. Furthermore, the part of the result and discussion sections were re-arranged.

**10. Generally, in the results and discussion, more description on the novelty of this study is needed. What are the new findings from this study compared to what already known, and what the significance and implication of the new findings for others?**

Answer: Thank you for your suggestion. In revised manuscript, we put results and discussion together. Moreover, more description on the novelty and implication of this study have been added in conclusion. Line 550-554.

This size distribution of AAs can help understand its transformation and fate in the atmosphere. However, detailed information on this topic is limited to a few studies and very variable results for the size-segregated concentrations and mole composition of atmospheric combined AAs have been observed in previous studies (Filippo et al. 2014; Scalabrin et al., 2012; Matsumoto and Uematsu, 2005; Barbaro et al., 2015). The factors controlling this large difference between fine and coarse particle are still unclear. Thus, verification of the different types, concentrations, origin and atmospheric processes of AAs distribution along the different air particle sizes is important and meaningful.

This study presents the first isotopic evidence that the sources of AAs for fine and coarse aerosol particles may be similar, all of which were influenced by biomass burning, soil, and plant sources. It is therefore that the huge difference in the concentrations and mol% compositions of THAAs between fine and coarse particles observed in this study is closely relevant to the degradation processes of AAs in aerosols.

Although the oxidation, nitrification and oligomerization processes of protein substances in the atmosphere have been widely reported in previous studies (Liu et al., 2017; Wang et al., 2019; Song et al., 2017; Haan et al., 2009), but these abiotic photochemical aging processes that occur between fine particles and coarse particles have not been compared. In this study, the difference in $\delta^{15}N$ values of Source-AA (Gly, Ser, Phe and Lys) and total hydrolysable amino acids ($\delta^{15}N_{THAA}$) between coarse particles and fine particles was relatively small (Fig. 10). The average offset of $\delta^{15}N_{THAA}$ value between fine and coarse particles was lower than 1.5‰ (Fig. 11a). These results appear to contrast with what one might expect for AAs in either sizes particles undergo particularly more photochemical transformation than the other.

On the contrary, the degradation of atmospheric AAs in aerosols is rarely investigated, except for one study on marine aerosols by Wedyan and Preston (2008). It is still unknown that whether bacterial degradation play a role in the levels and compositions of AAs in different particle sizes. This is the first report of using degradation marker (DI) to investigate the degradation state of aerosol particles. Both composition profiles of HAA and concentrations of THAAs in aerosols are showed to be closely related to DI. And fine particles had significantly higher DI values than that of coarse particles ($p<0.05$) (Fig. 7a), suggesting the degradation degree of amino acids in coarse particles is higher than that in fine particles.

Combining new compound-specific nitrogen isotope tool ($\delta^{15}$N-HAA) and effective bacterial heterotrophy indicator ($\sum V$) , "scattered" characteristic of $\delta^{15}$N distribution in Tr-AA and higher $\sum V$ values were observed in coarse particles in this study, which firstly provide evidence that the stronger degradation state the found in coarse particles are coupled with more bacterial heterotrophic resynthesis occurred in coarse particles. In conclusion, the difference in the THAA concentration and mol% composition distribution between fine and coarse particles may be related to AAs in coarse particles have stronger bacterial degradation state than those in fine particles.

Moreover, DI values in coarse aerosol particles were significant increased ($p<0.05$) but the $\sum V$ value was significantly decreased ($p < 0.05$) during rain events, suggesting more fresh AAs in coarse particles were released by droplets and it highly depends on the amounts and intensity of the rainfall.

This study firstly suggests the potentially significant role of bacterial degradation processes in concentration and composition of protein distribution in size-segregated aerosol particles. Since the degradation state of airborne protein distribution along size-segregated particles is closely linked to its biological availability, ecological processes and plant nutrition after deposition, further studies of quantitative assessment of this biological related process in aerosols should be conducted.

All changes can be tracked in the revised manuscript. Thank you very much again.

Yours sincerely,

[revised manuscript text omitted]

---

## Author Response (AR2)

**Dear editor:**

**Thank you for your letter and comments concerning our manuscript acp-2020-534 entitled "Measurement report: Hydrolyzed Amino acids in fine and coarse atmospheric aerosol in Nanchang, China: concentrations, compositions, sources and possible bacterial degradation state". Those comments are all valuable and very helpful improving our paper. We accept your suggestions and revise the manuscript according to the points as follows.**

Comments to the Author:
Authors,

Thank you for taking the time to put detailed work into the responses to the referees' comments. You saw the last round of comments by the final two referees. Based on their input, I'm happy for the manuscript to proceed to publication after a few minor additional edits.

Referee #2 brought up a good point about the correlation of precipitation to AA data, given the coarse nature of the sampling (~24-hr intervals). Given this, I think it would be useful to edit the wording in places to refer to days or samples and not precipitation "events," which are much shorter in nature. I suggest adding the following minor edits (line numbers refer to tracked version):

- Line 543: Define "higher precipitation" here (From Table S4 - looks 1 mm or more per day?)

- Line 40: This is an example of where I suggest changing from "events" to ~"days on which precipitation fell."

- Line 41: Given the sampling frequency, this is statement is a bit guess-work. I would change to "likely release" instead of "mainly released"

- Please also look through section 3.6 and change mentions of precipitation "events" to similar wording above about 'days on which precipitation fell', etc.

Thank you again for your submission to ACP and for your patience through this slower than normal process. The COVID situation has delayed my response time, for which I apologize.

Congratulations and best regards,

Alex Huffman

**Answer:** Thank you for your suggestion. We have noticed the difference between the precipitation events and rainy days. According to coarse nature of the sampling (~24-hr intervals), our data may reflect the AA variation between rainy days and dry days rather than the changes of AAs in precipitation events because the time of events are shorter. So, we have edited the wording in the abstract and section 3.6 as your suggestion.

**- Line 543: Define "higher precipitation" here (From Table S4 - looks 1 mm or more per day?**
**Answer:** Thank you for your suggestion. Here, we have defined the higher precipitation. The higher precipitation refers to the daily precipitation amount was above 1mm and the hourly rainfall amount was above 0.2mm (line 517-518 in tracked version).

**- Line 40: This is an example of where I suggest changing from "events" to ~"days on which precipitation fell."**
**Answer:** Thank you for your suggestion. We have corrected (line 35 in tracked version).

**- Line 41: Given the sampling frequency, this is statement is a bit guess-work. I would change to "likely release" instead of "mainly released"**
**Answer:** Thank you for your suggestion. "mainly" was changed to "likely" (line 36 in tracked version).

**- Please also look through section 3.6 and change mentions of precipitation "events" to similar wording above about 'days on which precipitation fell', etc.**
**Answer:** Thank you for your suggestion. "precipitation events" in the section 3.6 were all changed to "rainy days" or "days on which precipitation fell".

All changes can be tracked in the revised manuscript. Thank you very much again.

Yours sincerely,

[revised manuscript text omitted]